# HesScale: Scalable Computation of Hessian Diagonals

## Abstract

Second-order optimization uses curvature information about the objective function, which can help in faster convergence. However, such methods typically require expensive computation of the Hessian matrix, preventing their usage in a scalable way. The absence of efficient ways of computation drove the most widely used methods to focus on first-order approximations that do not capture the curvature information. In this paper, we develop *HesScale*, a scalable approach to approximating the diagonal of the Hessian matrix, to incorporate second-order information in a computationally efficient manner. We show that HesScale has the same computational complexity as backpropagation. Our results on supervised classification show that HesScale achieves high approximation accuracy, allowing for scalable and efficient second-order optimization.[1]

## 1 Introduction

First-order optimization offers a cheap and efficient way of performing local progress in optimization problems by using gradient information. However, their performance suffers from instability or slow progress when used in ill-conditioned landscapes. Such a problem is present because first-order methods do not capture curvature information which causes two interrelated issues. First, the updates in first-order have incorrect units (Duchi et al. 2011), which creates a scaling issue. Second, first-order methods lack parameterization invariance (Martens 2020) in contrast to second-order methods such as natural gradient (Amari 1998) or Newton-Raphson methods. Therefore, some first-order normalization methods were developed to address the invariance problem (Ba et al. 2016, Ioffe & Szegedy 2015, Salimans & Kingma 2016). On the other hand, some recent adaptive step-size methods try to alleviate the scaling issue by using gradient information for first-order curvature approximation (Luo et al. 2019, Duchi et al. 2011, Zeiler 2012, Reddi et al. 2018, Kingma & Ba 2015, Tran & Phong 2019, Tieleman et al. 2012). Specifically, such methods use the empirical Fisher diagonals heuristic by maintaining a moving average of the squared gradients to approximate the diagonal of the Fisher information matrix. Despite the huge adoption of such methods due to their scalability, they use inaccurate approximations. Kunstner et al. (2019) showed that the empirical Fisher does not generally capture curvature information and might have undesirable effects. They argued that the empirical Fisher approximates the Fisher or the Hessian matrices only under strong assumptions that are unlikely to be met in practice. Moreover, Wilson et al. (2017) presented a counterexample where the adaptive step-size methods are unable to reduce the error compared to non-adaptive counterparts such as stochastic gradient descent.

Although second-order optimization can speed up the training process by using the geometry of the landscape, its adoption is minimal compared to first-order methods. The exact natural gradient or Newton-Raphson methods require the computation, storage, and inversion of the Fisher information or the Hessian matrices, making them computationally prohibitive in large-scale tasks. Accordingly, many popular second-order methods attempt to approximate less expensively. For example, a type of truncated-Newton method called Hessian-free methods (Martens 2010) exploits the fact that the Hessian-vector product is cheap (Bekas et al. 2007) and uses the iterative conjugate gradient method to perform an update. However, such methods might require many iterations per update or some tricks to achieve stability, adding computational overhead (Martens & Sutskever 2011).

---

[1]Code will be available.

Some variations try to approximate only the diagonals of the Hessian matrix using stochastic estimation with matrix-free computations (Chapelle & Erhan 2011, Martens et al. 2012, Yao et al. 2021). Other methods impose probabilistic modeling assumptions and estimate a block diagonal Fisher information matrix (Martens & Grosse 2015, Botev et al. 2017). Such methods are invariant to reparametrization but are computationally expensive since they need to perform matrix inversion for each block.

Deterministic diagonal approximations to the Hessian (LeCun et al. 1990, Becker & Lecun 1989) provide some curvature information and are efficient to compute. Specifically, they can be implemented to be as efficient as first-order methods. We view this category of approximation methods as *scalable second-order methods*. In neural networks, curvature backpropagation (Becker & Lecun 1989) can be used to backpropagate the curvature vector. We distinguish this efficient method from other expensive methods (e.g., Mizutani & Dreyfus 2008, Botev et al. 2017) that backpropagate the full Hessian matrices. Although these diagonal methods show a promising direction for scalable second-order optimization, the approximation quality is sometimes poor with objectives such as cross-entropy (Martens et al. 2012). A scalable second-order method with high quality approximation is still needed.

In this paper, we present *HesScale*, a high-quality approximation method for the Hessian diagonals. Our method is also scalable and has little memory requirement with linear computational complexity while maintaining high approximation accuracy.

## 2 BACKGROUND

In this section, we describe the Hessian matrix for neural networks and some existing methods for estimating it. Generally, Hessian matrices can be computed for any scalar-valued function that are twice differentiable. If $f : \mathbb{R}^n \to \mathbb{R}$ is such a function, then for its argument $\boldsymbol{\psi} \in \mathbb{R}^n$, the Hessian matrix $\boldsymbol{H} \in \mathbb{R}^{n \times n}$ of $f$ with respect to $\boldsymbol{\psi}$ is given by $H_{i,j} = \partial^2 f(\boldsymbol{\psi}) / \partial \psi_i \partial \psi_j$. Here, the $i$th element of a vector $\boldsymbol{v}$ is denoted by $v_i$, and the element at the $i$th row and $j$th column of a matrix $\boldsymbol{M}$ is denoted by $M_{i,j}$. When the need for computing the Hessian matrix arises for optimization in deep learning, the function $f$ is typically the objective function, and the vector $\boldsymbol{\psi}$ is commonly the weight vector of a neural network. Computing and storing an $n \times n$ matrix, where $n$ is the number of weights in a neural network, is expensive. Therefore, many methods exist for approximating the Hessian matrix or parts of it with less memory footprint, computational requirement, or both. A common technique is to utilize the structure of the function to reduce the computations needed. For example, assuming that connections from a certain layer do not affect other layers in a neural network allows one to approximate a block diagonal Hessian. The computation further simplifies when we have piece-wise linear activation functions (e.g., ReLU), which result in a *Generalized Gauss-Newton* (GGN) (Schraudolph 2002) approximation that is equivalent to the block diagonal Hessian matrix with linear activation functions. The GGN matrix is more favored in second-order optimization since it is positive semi-definite. However, computing a block diagonal matrix is still demanding.

Many approximation methods were developed to reduce the storage and computation requirements of the GGN matrix. For example, under probabilistic modeling assumptions, the *Kronecker-factored Approximate Curvature* (KFAC) method (Martens & Grosse 2015) writes the GGN matrix $\boldsymbol{G}$ as a Kronecker product of two matrices of smaller sizes as: $\boldsymbol{G} = \boldsymbol{A} \otimes \boldsymbol{B}$, where $\boldsymbol{A} = \mathbb{E}[\boldsymbol{h}\boldsymbol{h}^\top]$, $\boldsymbol{B} = \mathbb{E}[\boldsymbol{g}\boldsymbol{g}^\top]$, $\boldsymbol{h}$ is the activation output vector, and $\boldsymbol{g}$ is the gradient of the loss with respect to the activation input vector. The $\boldsymbol{A}$ and $\boldsymbol{B}$ matrices can be estimated by Monte Carlo sampling and an exponential moving average. KFAC is more efficient when used in optimization since it requires inverting only the small matrices using the Kronecker-product property $(\boldsymbol{A} \otimes \boldsymbol{B})^{-1} = \boldsymbol{A}^{-1} \otimes \boldsymbol{B}^{-1}$. However, KFAC is still expensive due to the storage of the block diagonal matrices and computation of Kronecker product, which prevent it from being used as a scalable method.

Computing the Hessian diagonals can provide some curvature information with relatively less computation. However, it has been shown that the exact computation for diagonals of the Hessian typically has quadratic complexity with the unlikely existence of algorithms that can compute the exact diagonals with less than quadratic complexity (Martens et al. 2012). Some stochastic methods provide a way to compute unbiased estimates of the exact Hessian diagonals. For example, the AdaHessian (Yao et al. 2021) algorithm uses the Hutchinson's estimator $\text{diag}(\boldsymbol{H}) = E[\boldsymbol{z} \circ (\boldsymbol{H}\boldsymbol{z})]$, where $\boldsymbol{z}$ is a multivariate random variable with a Rademacher distribution and the expectation can

be estimated using Monte Carlo sampling with an exponential moving average. Similarly, the GGN-MC method (Dangel et al. 2020) uses the relationship between the Fisher information matrix and the Hessian matrix under probabilistic modeling assumptions to have an MC approximation of the diagonal of the GGN matrix. Although these stochastic approximation methods are scalable due to linear or $O(n)$ computational and memory complexity, they suffer from low approximation quality, improving which requires many sampling and factors of additional computations.

## 3 THE PROPOSED HESSCALE METHOD

In this section, we present our method for approximating the diagonal of the Hessian at each layer in feed-forward networks, where a backpropagation rule is used to utilize the Hessian of previous layers. We present the derivation of the backpropagation rule for fully connected and convolutional neural networks in supervised learning. Similar derivation for fully connected networks with mean squared error is presented before (LeCun et al. 1990, Becker & Lecun 1989). However, we use the exact diagonals of the Hessian matrix at the last layer with some non-linear and non-element-wise output activations such as softmax and show that it can still be computed in linear computational complexity. We show the derivation for Hessian diagonals for fully connected networks in the following and provide the derivation for the convolutional neural networks in Appendix B.

We use the supervised classification setting where there is a collection of data examples. These data examples are generated from some *target function* $f^*$ mapping the input $\boldsymbol{x}$ to the output $y$, where the $k$-th input-output pair is $(\boldsymbol{x}_k, y_k)$. In this task, the *learner* is required to predict the output class $y \in \{1, 2, ..., m\}$ given the input vector $\boldsymbol{x} \in \mathbb{R}^d$ by estimating the target function $f^*$. The performance is measured with the cross-entropy loss, $\mathcal{L}(\boldsymbol{p}, \boldsymbol{q}) = -\sum_{i=1}^m p_i \log q_i$, where $\boldsymbol{p} \in \mathbb{R}^m$ is the vector of the target one-hot encoded class and $\boldsymbol{q} \in \mathbb{R}^m$ is the predicted output. The learner is required to reduce the cross-entropy by matching the target class.

Consider a neural network with $L$ layers that outputs the predicted output $\boldsymbol{q}$. The neural network is parametrized by the set of weights $\{\boldsymbol{W}_1, ..., \boldsymbol{W}_L\}$, where $\boldsymbol{W}_l$ is the weight matrix at the $l$-th layer, and its element at the $i$th row and the $j$th column is denoted by $W_{l,i,j}$. During learning, the parameters of the neural network are changed to reduce the loss. At each layer $l$, we get the activation output $\boldsymbol{h}_l$ by applying the activation function $\boldsymbol{\sigma}$ to the activation input $\boldsymbol{a}_l$: $\boldsymbol{h}_l = \boldsymbol{\sigma}(\boldsymbol{a}_l)$. We simplify notations by defining $\boldsymbol{h}_0 \doteq \boldsymbol{x}$. The activation output $\boldsymbol{h}_l$ is then multiplied by the weight matrix $\boldsymbol{W}_{l+1}$ of layer $l + 1$ to produce the next activation input: $a_{l+1,i} = \sum_{j=1}^{|\boldsymbol{h}_l|} W_{l+1,i,j} h_{l,j}$. We assume here that the activation function is element-wise activation for all layers except for the final layer $L$, where it becomes the softmax function. The backpropagation equations for the described network are given as follows Rumelhart et al. (1986):

$$\frac{\partial \mathcal{L}}{\partial a_{l,i}} = \sum_{k=1}^{|\boldsymbol{a}_{l+1}|} \frac{\partial \mathcal{L}}{\partial a_{l+1,k}} \frac{\partial a_{l+1,k}}{\partial h_{l,i}} \frac{\partial h_{l,i}}{\partial a_{l,i}} = \sigma'(a_{l,i}) \sum_{k=1}^{|\boldsymbol{a}_{l+1}|} \frac{\partial \mathcal{L}}{\partial a_{l+1,k}} W_{l+1,k,i}, \tag{1}$$

$$\frac{\partial \mathcal{L}}{\partial W_{l,i,j}} = \frac{\partial \mathcal{L}}{\partial a_{l,i}} \frac{\partial a_{l,i}}{\partial W_{l,i,j}} = \frac{\partial \mathcal{L}}{\partial a_{l,i}} h_{l-1,j}. \tag{2}$$

In the following, we write the equations for the exact Hessian diagonals with respect to weights $\partial^2 \mathcal{L}/\partial W_{l,i,j}^2$, which requires the calculation of $\partial^2 \mathcal{L}/\partial a_{l,i}^2$ first:

$$\frac{\partial^2 \mathcal{L}}{\partial a_{l,i}^2} = \frac{\partial}{\partial a_{l,i}} \left( \sigma'(a_{l,i}) \sum_{k=1}^{|\boldsymbol{a}_{l+1}|} \frac{\partial \mathcal{L}}{\partial a_{l+1,k}} W_{l+1,k,i} \right)$$

$$= \sigma'(a_{l,i}) \sum_{k=1}^{|\boldsymbol{a}_{l+1}|} \sum_{p=1}^{|\boldsymbol{a}_{l+1}|} \frac{\partial^2 \mathcal{L}}{\partial a_{l+1,k} \partial a_{l+1,p}} \frac{\partial a_{l+1,p}}{\partial a_{l,i}} W_{l+1,k,i} + \sigma''(a_{l,i}) \sum_{k=1}^{|\boldsymbol{a}_{l+1}|} \frac{\partial \mathcal{L}}{\partial a_{l+1,k}} W_{l+1,k,i}$$

$$= \sigma'(a_{l,i})^2 \sum_{k=1}^{|\boldsymbol{a}_{l+1}|} \sum_{p=1}^{|\boldsymbol{a}_{l+1}|} \frac{\partial^2 \mathcal{L}}{\partial a_{l+1,k} \partial a_{l+1,p}} W_{l+1,p,i} W_{l+1,k,i} + \sigma''(a_{l,i}) \sum_{k=1}^{|\boldsymbol{a}_{l+1}|} \frac{\partial \mathcal{L}}{\partial a_{l+1,k}} W_{l+1,k,i},$$

$$\frac{\partial^2 \mathcal{L}}{\partial W_{l,i,j}^2} = \frac{\partial}{\partial W_{l,i,j}} \left( \frac{\partial \mathcal{L}}{\partial a_{l,i}} h_{l-1,j} \right) = \frac{\partial}{\partial a_{l,i}} \left( \frac{\partial \mathcal{L}}{\partial a_{l,i}} \right) \frac{\partial a_{l,i}}{\partial W_{l,i,j}} h_{l-1,j} = \frac{\partial^2 \mathcal{L}}{\partial a_{l,i}^2} h_{l-1,j}^2. \tag{3}$$

Since, the calculation of $\partial^2 \mathcal{L}/\partial a_{l,i}^2$ depends on the off-diagonal terms, the computation complexity becomes quadratic. Following Becker and Lecun (1989), we approximate the Hessian diagonals by ignoring the off-diagonal terms, which leads to a backpropagation rule with linear computational complexity for our estimates $\widehat{\frac{\partial^2 \mathcal{L}}{\partial W_{l,i,j}^2}}$ and $\widehat{\frac{\partial^2 \mathcal{L}}{\partial a_{l,i}^2}}$:

$$\widehat{\frac{\partial^2 \mathcal{L}}{\partial a_{l,i}^2}} \doteq \sigma'(a_{l,i})^2 \sum_{k=1}^{|\boldsymbol{a}_{l+1}|} \widehat{\frac{\partial^2 \mathcal{L}}{\partial a_{l+1,k}^2}} W_{l+1,k,i}^2 + \sigma''(a_{l,i}) \sum_{k=1}^{|\boldsymbol{a}_{l+1}|} \frac{\partial \mathcal{L}}{\partial a_{l+1,k}} W_{l+1,k,i}, \qquad (4)$$

$$\widehat{\frac{\partial^2 \mathcal{L}}{\partial W_{l,i,j}^2}} \doteq \widehat{\frac{\partial^2 \mathcal{L}}{\partial a_{l,i}^2}} h_{l-1,j}^2. \qquad (5)$$

However, for the last layer, we use the exact Hessian diagonals $\widehat{\frac{\partial^2 \mathcal{L}}{\partial a_{L,i}^2}} \doteq \frac{\partial^2 \mathcal{L}}{\partial a_{L,i}^2}$ since it can be computed in $O(n)$ for the softmax activation function and the cross-entropy loss. More precisely, the exact Hessian diagonals for cross-entropy loss with softmax is simply $\boldsymbol{q} - \boldsymbol{q} \circ \boldsymbol{q}$, where $\boldsymbol{q}$ is the predicted probability vector and $\circ$ denotes element-wise multiplication. We found empirically that this small change makes a large difference in the approximation quality, as shown in Fig. 1a. Hence, unlike Becker and Lecun (1989) who use a Hessian diagonal approximation of the last layer by Eq. 4, we use the exact values directly to achieve more approximation accuracy. We call this method for Hessian diagonal approximation *HesScale* and provide its pseudocode for supervised classification in Algorithm 1. HesScale is not specific to cross-entropy loss as the exact Hessian diagonals can

---

**Algorithm 1** HesScale: Computing Hessian diagonals of a neural network layer in classification

---

**Require:** Neural network $f$ and a layer number $l$
**Require:** First and second order information $\widehat{\frac{\partial \mathcal{L}}{\partial a_{l+1,i}}}$ and $\widehat{\frac{\partial^2 \mathcal{L}}{\partial a_{l+1,i,j}^2}}$, unless $l = L$
**Require:** Input-output pair $(\boldsymbol{x}, y)$
  Set loss function $\mathcal{L}$ to cross-entropy loss
  Compute preference vector $\boldsymbol{a}_L \leftarrow f(\boldsymbol{x})$ and target one-hot-encoded vector $\boldsymbol{p} \leftarrow \texttt{onehot}(y)$
  Compute the predicted probability vector $\boldsymbol{q} \leftarrow \boldsymbol{\sigma}(\boldsymbol{a}_L)$ using softmax function $\boldsymbol{\sigma}$
  Compute the error $\mathcal{L}(\boldsymbol{p}, \boldsymbol{q})$
  **if** $l = L$ **then**                  $\triangleright$ Computing Hessian diagonals exactly for the last layer
      Compute $\frac{\partial \mathcal{L}}{\partial \boldsymbol{a}_L} \leftarrow \boldsymbol{q} - \boldsymbol{p}$                 $\triangleright$ $\frac{\partial \mathcal{L}}{\partial \boldsymbol{a}_L}$ consists of elements $\frac{\partial \mathcal{L}}{\partial a_{L,i}}$
      Compute $\frac{\partial \mathcal{L}}{\partial \boldsymbol{W}_L}$ using Eq. 2           $\triangleright$ $\frac{\partial \mathcal{L}}{\partial \boldsymbol{W}_L}$ consists of elements $\frac{\partial \mathcal{L}}{\partial W_{L,i,j}}$
      $\widehat{\frac{\partial^2 \mathcal{L}}{\partial \boldsymbol{a}_L^2}} \leftarrow \boldsymbol{q} - \boldsymbol{q} \circ \boldsymbol{q}$            $\triangleright$ $\widehat{\frac{\partial^2 \mathcal{L}}{\partial \boldsymbol{a}_L^2}}$ consists of elements $\widehat{\frac{\partial^2 \mathcal{L}}{\partial a_{L,i}^2}}$
      Compute $\widehat{\frac{\partial^2 \mathcal{L}}{\partial \boldsymbol{W}_L^2}}$ using Eq. 5         $\triangleright$ $\widehat{\frac{\partial^2 \mathcal{L}}{\partial \boldsymbol{W}_L^2}}$ consists of elements $\widehat{\frac{\partial^2 \mathcal{L}}{\partial W_{L,i,j}^2}}$
  **else if** $l \neq L$ **then**
      Compute $\frac{\partial \mathcal{L}}{\partial \boldsymbol{a}_l}$ and $\partial \mathcal{L}/\partial \boldsymbol{W}_l$ using Eq. 1 and Eq. 2
      Compute $\widehat{\frac{\partial^2 \mathcal{L}}{\partial \boldsymbol{a}_l^2}}$ and $\widehat{\frac{\partial^2 \mathcal{L}}{\partial \boldsymbol{W}_l^2}}$ using Eq. 4 and Eq. 5
  **end if**
  **return** $\frac{\partial \mathcal{L}}{\partial \boldsymbol{W}_l}$, $\widehat{\frac{\partial^2 \mathcal{L}}{\partial \boldsymbol{W}_l^2}}$, $\frac{\partial \mathcal{L}}{\partial \boldsymbol{a}_l}$, and $\widehat{\frac{\partial^2 \mathcal{L}}{\partial \boldsymbol{a}_l^2}}$

---

be calculated in $O(n)$ for some other widely used loss functions as well. We show this property for negative log-likelihood function with Gaussian and softmax distributions in Appendix A. The computations can be reduced further using a linear approximation for the activation functions (by dropping the second term in Eq. 4), which corresponds to an approximation of the GGN matrix. We call this variation of our method *HesScaleGN*.

Based on HesScale, we make a stable optimizer, which we call *AdaHesScale*, given in Algorithm 2. We use the same style introduced in Adam (Kingma & Ba 2015), using the squared diagonal approximation instead of the squared gradients to update the moving average. Moreover, we introduce another optimizer based on HesScaleGN, which we call *AdaHesScaleGN*. We refer the reader to the convergence proof for methods with Hessian diagonals, which was presented by Yao et al. (2021).

---

**Algorithm 2** AdaHesScale for optimization

---

**Require:** Neural network $f$ with weights $\{\boldsymbol{W}_1, ..., \boldsymbol{W}_L\}$ and a dataset $\mathcal{D}$
**Require:** Small number $\epsilon \leftarrow 10^{-8}$
**Require:** Exponential decay rates $\beta_1, \beta_2 \in [0, 1)$
**Require:** step size $\alpha$
**Require:** Initialize $\{\boldsymbol{W}_1, ..., \boldsymbol{W}_L\}$
  Initialize time step $t \leftarrow 0$.
  **for** $l$ in $\{L, L-1, ..., 1\}$ **do**         ▷ Set exponential moving averages at time step 0 to zero
     $\boldsymbol{M}_l \leftarrow 0; \quad \boldsymbol{V}_l \leftarrow 0$                       ▷ Same size as $\boldsymbol{W}_l$
  **end for**
  **for** $(\boldsymbol{x}, y)$ in $\mathcal{D}$ **do**
    $t \leftarrow t + 1$
    $\boldsymbol{r}_{L+1} \leftarrow \boldsymbol{s}_{L+1} \leftarrow \emptyset$         ▷ $\boldsymbol{r}_l$ and $\boldsymbol{s}_l$ stand for $\frac{\partial \mathcal{L}}{\partial \boldsymbol{a}_l}$ and $\widehat{\frac{\partial^2 \mathcal{L}}{\partial \boldsymbol{a}_l^2}}$, respectively
    **for** $l$ in $\{L, L-1, ..., 1\}$ **do**
      $\boldsymbol{F}_l, \boldsymbol{S}_l, \boldsymbol{r}_l, \boldsymbol{s}_l \leftarrow \texttt{HesScale}(f, \boldsymbol{x}, y, l, \boldsymbol{r}_{l+1}, \boldsymbol{s}_{l+1})$.      ▷ Check Algorithm 1
      $\boldsymbol{M}_l \leftarrow \beta_1 \boldsymbol{M}_l + (1 - \beta_1) \boldsymbol{F}_l$           ▷ $\boldsymbol{F}_l$ stands for $\frac{\partial \mathcal{L}}{\partial \boldsymbol{W}_l}$
      $\boldsymbol{V}_l \leftarrow \beta_2 \boldsymbol{V}_l + (1 - \beta_2) \boldsymbol{S}_l^2$           ▷ $\boldsymbol{S}_l$ stands for $\widehat{\frac{\partial^2 \mathcal{L}}{\partial \boldsymbol{W}_l^2}}$
      $\hat{\boldsymbol{M}}_l \leftarrow \boldsymbol{M}_l/(1 - \beta_1^t)$           ▷ Bias-corrected estimate for $\boldsymbol{F}_l$
      $\hat{\boldsymbol{V}}_l \leftarrow \boldsymbol{V}_l/(1 - \beta_2^t)$            ▷ Bias-corrected estimate for $\boldsymbol{S}_l$
      $\boldsymbol{W}_l \leftarrow \boldsymbol{W}_l - \alpha \hat{\boldsymbol{M}}_l \oslash (\hat{\boldsymbol{V}}_l + \epsilon)^{\circ \frac{1}{2}}$        ▷ $\oslash$ is element-wise division
                                              ▷ $\boldsymbol{A}^{\circ \frac{1}{2}}$ is element-wise square root of $\boldsymbol{A}$
    **end for**
  **end for**

---

## 4   APPROXIMATION QUALITY & SCALABILITY OF HESSCALE

In this section, we evaluate HesScale for its approximation quality and computational cost and compare it with other methods. These measures constitute the criteria we look for in scalable and efficient methods. For our experiments, we implemented HesScale using the *BackPack* framework (Dangel et al. 2020), which allows easy implementation of backpropagation of statistics other than the gradient.

We start by studying the approximation quality of Hessian diagonals compared to the true values. To measure the approximation quality of the Hessian diagonals for different methods, we use the $L^1$ distance between the exact Hessian diagonals and their approximations. Our task here is supervised classification, and data examples are randomly generated. We used a network of three hidden layers with *tanh* activations, each containing 16 units. The network weights and biases are initialized randomly. The network has six inputs and ten outputs. For each example pair, we compute the exact Hessian diagonals for each layer and their approximations from each method. All layers' errors are summed and averaged over 1000 data examples for each method. In this experiment, we used 40 different initializations for the network weights, shown as colored dots in Fig. 1a. Each point represents the summed error over network layers, averaged over 1000 examples for each different initialization. In this figure, we show the average error incurred by each method normalized by the average error incurred by HesScale. Any approximation that incurs an averaged error above 1 has a worse approximation than HesScale, and any approximation with an error less than 1 has a better approximation than HesScale. Moreover, we show the layer-wise error for each method in Fig. 1b.

Different Hessian diagonal approximations are considered for comparison with HesScale. We included several deterministic and stochastic approximations for the Hessian diagonals. We also include the approximation of the Fisher Information Matrix done by squaring the gradients and denoted by $g^2$, which is highly adopted by many first-order methods (e.g., Kingma and Ba, 2015). We compare HesScale with three stochastic approximation methods: AdaHessian (Yao et al. 2021), Kronecker-factored approximate curvature (KFAC) (Martens & Grosse 2015), and the Monte-Carlo (MC) estimate of the GGN matrix (GGN-MC) (Dangel et al. 2020). We also compare HesScale with two deterministic approximation methods: the diagonals of the exact GGN matrix (Schraudolph 2002) ($\mathrm{diag}(\boldsymbol{G})$) and the diagonal approximation by Becker and Lecun (1989) (BL89). In

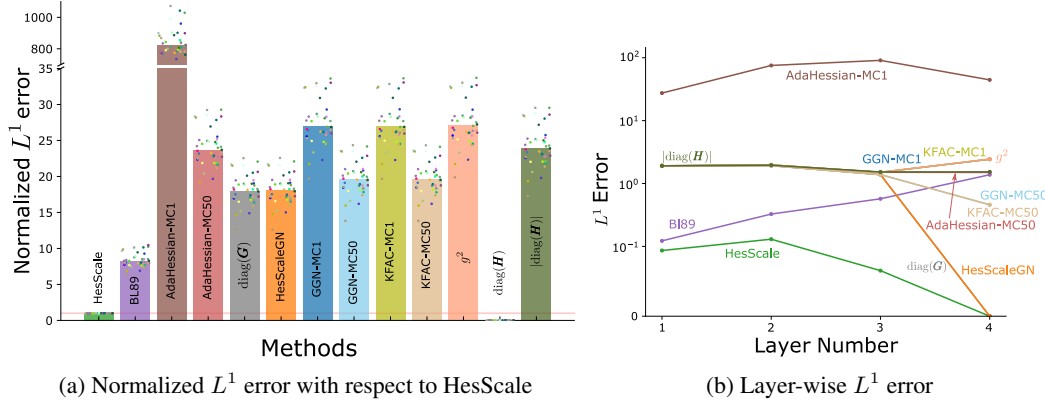

(a) Normalized $L^1$ error with respect to HesScale

(b) Layer-wise $L^1$ error

Figure 1: The averaged error for each method is normalized by the averaged error incurred by HesScale. We show 40 initialization points with the same colors across all methods. The norm of the vector of Hessian diagonals $|\text{diag}(\boldsymbol{H})|$ is shown as a reference.

KFAC, we extract the diagonals from the block diagonal matrix and show the approximation error averaged over 1 MC sample (KFAC-MC1) and over 50 MC samples (KFAC-MC50), both per each data example. Since AdaHessian and GGN-MC are already diagonal approximations, we use them directly and show the error with 1 MC sample (GGN-MC1 & AdaHessian-MC1) and with 50 MC samples (GGN-MC50 & AdaHessian-MC50). We refer the reader to Appendix C for an additional experiment with MNIST data points.

HesScale provides a better approximation than the other deterministic and stochastic methods. For stochastic methods, we use many MC samples to improve their approximation. However, their approximation quality is still poor. Methods approximating the GGN diagonals do not capture the complete Hessian information since the GGN and Hessian matrices are different when the activation functions are not piece-wise linear. Although these methods approximate the GGN diagonals, their approximation is significantly better than the AdaHessian approximation. And among the methods for approximating the GGN diagonals, HesScaleGN performs the best and is close to the exact GGN diagonals. This experiment clearly shows that HesScale achieves the best approximation quality compared to other stochastic and deterministic approximation methods.

Next, we perform another experiment to evaluate the computational cost of our optimizers. Our Hessian approximation methods and corresponding optimizers have linear computational complexity, which can be seen from Eq. 4 and Eq. 5. However, computing second-order information in optimizers still incurs extra computations compared to first-order optimizers, which may impact how the total computations scale with the number of parameters. Hence, we compare the computational cost of our optimizers with others for various numbers of parameters. More specifically, we measure the update time of each optimizer, which is the time needed to backpropagate first-order and second-order information and update the parameters.

We designed two experiments to study the computational cost of first-order and second-order optimizers. In the first experiment, we used a neural network with a single hidden layer. The network has 64 inputs and 512 hidden units with *tanh* activations. We study the increase in computational time when increasing the number of outputs exponentially, which roughly doubles the number of parameters. The set of values we use for the number of outputs is $\{2^4, 2^5, 2^6, 2^7, 2^8, 2^9\}$. The results of this experiment are shown in Fig. 2a. In the second experiment, we used a neural network with multi-layers, each containing 512 hidden units with *tanh* activations. The network has 64 inputs and 100 outputs. We study the increase in computational time when increasing the number of layers exponentially, which also roughly doubles the number of parameters. The set of values we use for the number of layers is $\{1, 2, 4, 8, 16, 32, 64, 128\}$. The results are shown in Fig. 2b. The points in Fig. 2a and Fig. 2b are averaged over 30 updates. The standard errors of the means of these points are smaller than the width of each line. On average, we notice that the cost of AdaHessian, AdaHesScale, and AdaHesScaleGN are three, two, and 1.25 times the cost of Adam, respectively.

It is clear that our methods are among the most computationally efficient approximation method for Hessian diagonals.

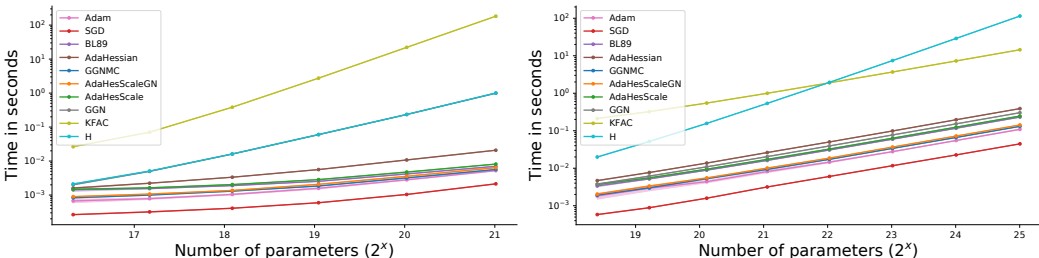

(a) Increasing number of outputs in a neural network     (b) Increasing number of layers in a neural network

Figure 2: The average computation time for each step of an update is shown for different optimizers. The computed update time is the time needed by each optimizer to backpropagate gradients or second-order information and to update the parameters of the network. GGN overlaps with H in (a).

## 5 EMPIRICAL PERFORMANCE OF HESSCALE IN OPTIMIZATION

In this section, we compare the performance of our optimizers—AdaHesScale and AdaHesScaleGN—with three second-order optimizers: BL89, GGNMC, and AdaHessian. We also include comparisons to two first-order methods: Adam and SGD. We exclude KFAC and the exact diagonals of the GGN matrix from our comparisons due to their prohibitive computations.

Our optimizers are evaluated in the supervised classification problem with a series of experiments using different architectures and three datasets: MNIST, CIFAR-10, and CIFAR-100. Instead of attempting to achieve state-of-the-art performance with specialized techniques and architectures, we follow the DeepOBS benchmarking work (Schneider et al. 2019) and compare the optimizers in their generic and pristine form using relatively simple networks. It allows us to perform a more fair comparison without extensively utilizing specialized knowledge for a particular task. In the first experiment, we use the MNIST-MLP task from DeepOBS. The images are flattened and used as inputs to a network of three fully connected layers (1000, 500, and 100 units) with *tanh* activations. We train each method for 100 epochs with a batch size of 128. We show the training plots in Fig. 7a with their corresponding sensitivity plots in Appendix D, Fig. 9a. In the second experiment, we use the CIFAR10-3C3D task from the DeepOBS benchmarking tasks. The network consists of three convolutional layers with *tanh* activations, each followed by max pooling. After that, two fully connected layers (512 and 256 units) with *tanh* activations are used. We train each method for 100 epochs with a batch size of 128. We show the training plots in Fig. 7b with their corresponding sensitivity plots in Fig. 9b. In the third experiment, we use the CIFAR100-3C-3D task from DeepOBS. The network is the same as the one used in the second task except for the activations are *ELU*. We train each method for 200 epochs with a batch size of 128. We show the training plots in Fig. 8b with their corresponding sensitivity plots in Fig. 10b. In the fourth experiment, we use the CIFAR100-ALL-CNN task from DeepOBS with the ALL-CNN-C network, which consists of 9 convolutional layers (Springenberg et al. 2015) with *ELU* activations. We use *tanh* and *ELU* instead of *ReLU*, which is used in DeepOBS, to differentiate between the performance of AdaHesScale and AdaHesScaleGN. We show the training plots in Fig. 8a with their corresponding sensitivity plots in Fig. 10a.

In the MNIST-MLP and CIFAR-10-3C3D experiments, we performed a hyperparameter search for each method to determine the best set of $\beta_1$, $\beta_2$, and $\alpha$. The range of $\beta_2$ is $\{0.99, 0.999, 0.9999\}$ and the range of $\beta_1$ is $\{0.0, 0.9\}$. The range of step size is selected for each method to create a convex curve. Our criterion was to find the best hyperparameter configuration for each method in the search space that minimizes the area under the validation loss curve. The performance of each method was averaged over 30 independent runs. Each independent run had the same initial representation for the algorithms used in an experiment. Using each method's best hyperparameter configuration on the validation set, we show the performance of each method against the time in seconds needed to complete the required number of epochs, which better depicts the computational efficiency of the

methods. Fig. 3a and Fig. 3b show these results on MNIST-MLP and CIFAR-10 tasks. Moreover, we show the sensitivity of each method to the step size in Fig. 5a and Fig. 5b.

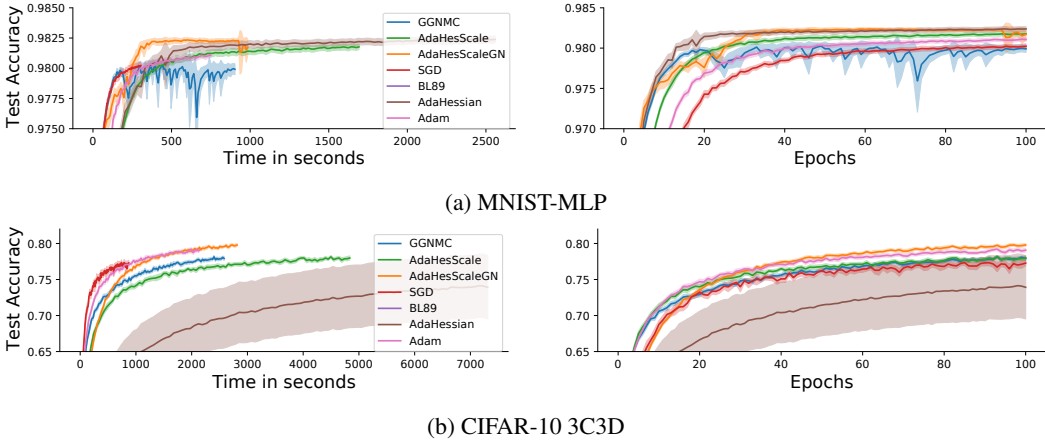

(a) MNIST-MLP

(b) CIFAR-10 3C3D

Figure 3: MNIST-MLP and CIFAR-10 3C3D classification tasks. Each method is trained for 100 epochs. We show the time taken by each algorithm in seconds (left) and we show the learning curves in the number of epochs (right). The performance of each method is averaged over 30 independent runs. The shaded area represents the standard error.

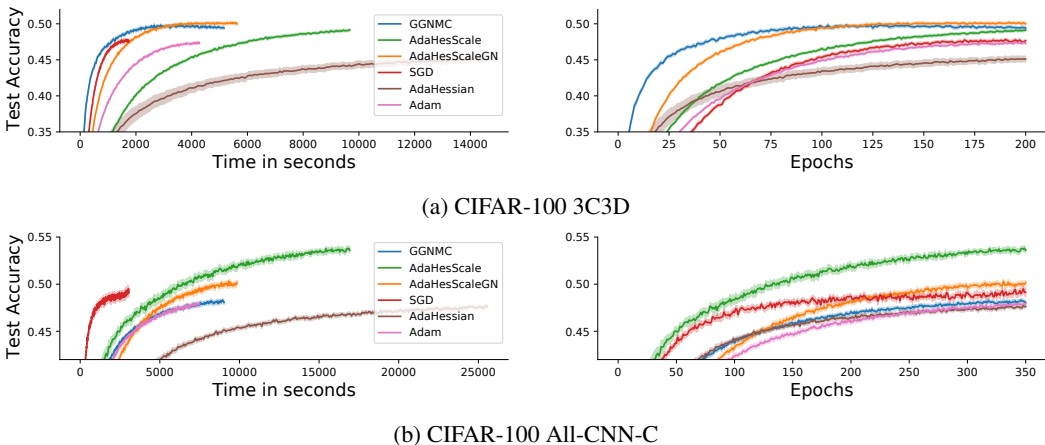

(a) CIFAR-100 3C3D

(b) CIFAR-100 All-CNN-C

Figure 4: CIFAR-100 3C3D and CIFAR-100 ALL-CNN classification tasks. Each method from the first task is trained for 200 epochs and each method from the second task is trained for 350 epochs. We show the time taken by each algorithm in seconds (left) and we show the learning curves in the number of epochs (right). The performance of each method is averaged over 30 independent runs. The shaded area represents the standard error.

In the CIFAR-100-ALL-CNN and CIFAR-100-3C3D experiments, we used the set of $\beta_1$ and $\beta_2$ that achieved the best robustness in the previous two tasks, which were 0.9 and 0.999 respectively. We did a hyperparameter search for each method to determine the best step size using the specified $\beta_1$ and $\beta_2$. The rest of the experimental details are the same as the first two experiments. Using each method's best hyperparameter configuration on the validation set, we show the performance of each method against the time in seconds needed to complete the required number of epochs. Fig. 4a and Fig. 4b show these results on CIFAR-100-ALL-CNN and CIFAR-100-3C3D tasks. We summarize the results in Appendix E.

Our results show that all optimizers except for BL89 performed well on the MNIST-MLP task. However, in CIFAR-10, CIFAR-100 3c3d, and CIFAR-100 ALL-CNN, we notice that AdaHessian performed worse than all methods except BL89. This result is aligned with AdaHessian's inability

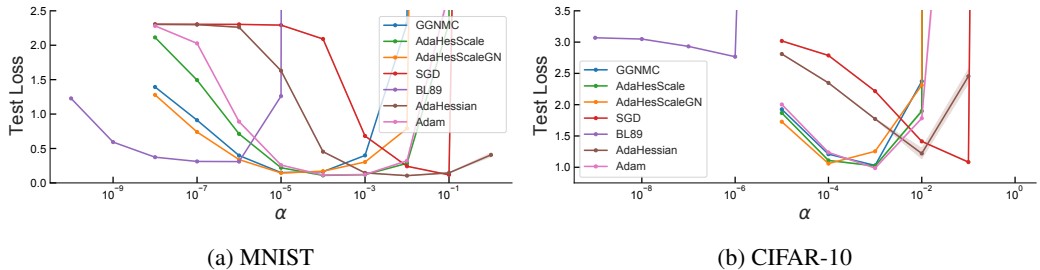

(a) MNIST           (b) CIFAR-10

Figure 5: Sensitivity of the step size for each method on MNIST-MLP and CIFAR-10 3C3D tasks. We select the best values of $\beta_1$ and $\beta_2$ for each step size $\alpha$.

to accurately approximate the Hessian diagonals, as shown in Fig. 1. Moreover, AdaHessian required more computational time compared to all methods, which is also reflected in Fig. 2. While being time-efficient, AdaHesScaleGN consistently outperformed all methods in CIFAR-10-3C3D and CIFAR-100-3C3D, and it outperformed all methods except AdaHesScale in CIFAR-100 ALL-CNN. This result is aligned with our methods' accurate approximation of Hessian diagonals. Our experiments indicate that incorporating HesScale and HesScaleGN approximations in optimization methods can be of significant performance advantage in both computation and accuracy. AdaHesScale and AdaHesScaleGN outperformed other optimizers likely due to their accurate approximation of the diagonals of the Hessian and GGN, respectively.

## 6 CONCLUSION

HesScale is a scalable and efficient second-order method for approximating the diagonals of the Hessian at every network layer. Our work is based on the previous work done by Becker and Lecun (1989). We performed a series of experiments to evaluate HesScale against other scalable algorithms in terms of computational cost and approximation accuracy. Moreover, we demonstrated how Hes-Scale can be used to build efficient second-order optimization methods. Our results showed that our methods provide a more accurate approximation and require small additional computations.

## 7 BROADER IMPACT

Second-order information is used in domains other than optimization. For example, some works alleviating catastrophic forgetting use a utility measure for the network's connections to protect them. Typically, an auxiliary loss is used between such connections, and their old values are weighted by their corresponding importance. Such methods (LeCun et al. 1990, Hassibi & Stork 1993, Dong et al. 2017, Kirkpatrick et al. 2017, Schwarz et al. 2018, Ritter et al. 2018) use the diagonal of the Fisher information matrix or the Hessian matrix as a utility measure for each weight. The quality of these algorithms depends heavily on the approximation quality of the second-order approximation. Second-order information can also be used in neural network pruning. Molchanov et al. (2019) showed that second-order approximation with the exact Hessian diagonals could closely represent the true measure of the utility of each weight.

The accurate and efficient approximation for the diagonals of the Hessian at each layer enables Hes-Scale to be used in many important lines of research. Using this second-order information provides a reliable measure of connection utility. Therefore, using HesScale in these types of problems can potentially improve the performance of neural network pruning methods and regularization-based catastrophic forgetting prevention methods.

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

# A  HESSIAN DIAGONALS OF THE LOG-LIKELIHOOD FUNCTION FOR TWO COMMON DISTRIBUTIONS

Here, we provide the diagonals of the Hessian matrix of functions involving the log-likelihood of two common distributions: a normal distribution and a categorical distribution with probabilities represented by a softmax function, which we refer to as a *softmax distribution*. We show that the exact computations of the diagonal can be computed with linear complexity since computing the diagonal elements does not depend on off-diagonals in these cases. In the following, we consider the softmax and normal distributions, and we write the exact Hessian diagonals in both cases.

## A.1  SOFTMAX DISTRIBUTION

Consider a cross-entropy function for a discrete probability distribution as $f \doteq -\sum_{i=1}^{|\boldsymbol{q}|} p_i \log q_i(\boldsymbol{\theta})$, where $\boldsymbol{q}$ is a probability vector that depends on a parameter vector $\boldsymbol{\theta}$, and $\boldsymbol{p}$ is a one-hot vector for the target class. For softmax distributions, $\boldsymbol{q}$ is parametrized by a softmax function $\boldsymbol{q} \doteq e^{\boldsymbol{\theta}} / \sum_{i=1}^{|\boldsymbol{q}|} e^{\theta_i}$. In this case, we can write the gradient of the cross-entropy function with respect to $\boldsymbol{\theta}$ as

$$\nabla_{\boldsymbol{\theta}} f(\boldsymbol{\theta}) = \boldsymbol{q} - \boldsymbol{p}.$$

Next, we write the exact diagonal elements of the Hessian matrix as follows:

$$\mathrm{diag}(\boldsymbol{H_\theta}) = \mathrm{diag}(\nabla_{\boldsymbol{\theta}}(\boldsymbol{q} - \boldsymbol{p})) = \boldsymbol{q} - \boldsymbol{q}^2,$$

where $\boldsymbol{q}^2$ denotes element-wise squaring of $\boldsymbol{q}$, and $\nabla$ operator applied to a vector denotes Jacobian. Computing the exact diagonals of the Hessian matrix depends only on vector operations, which means that we can compute it in $O(n)$. The cross-entropy loss is used with softmax distribution in many important tasks, such as supervised classification and discrete reinforcement learning control with parameterized policies (Chan et al. 2022).

## A.2  MULTIVARIATE NORMAL DISTRIBUTION WITH DIAGONAL COVARIANCE

For a multivariate normal distribution with diagonal covariance, the parameter vector $\boldsymbol{\theta}$ is determined by the mean-variance vector pair: $\boldsymbol{\theta} \doteq (\boldsymbol{\mu}, \boldsymbol{\sigma}^2)$. The log-likelihood of a random vector $\boldsymbol{x}$ drawn from this distribution can be written as

$$\log q(\boldsymbol{x}; \boldsymbol{\mu}, \boldsymbol{\sigma}^2) = -\frac{1}{2}(\boldsymbol{x} - \boldsymbol{\mu})^{\top} \boldsymbol{D}(\boldsymbol{\sigma}^2)^{-1}(\boldsymbol{x} - \boldsymbol{\mu}) - \frac{1}{2}\log(|\boldsymbol{D}(\boldsymbol{\sigma}^2)|) + c$$

$$= -\frac{1}{2}(\boldsymbol{x} - \boldsymbol{\mu})^{\top} \boldsymbol{D}(\boldsymbol{\sigma}^2)^{-1}(\boldsymbol{x} - \boldsymbol{\mu}) - \frac{1}{2}\log(\sum_{i=1}^{|\boldsymbol{\sigma}|} \sigma_i^2) + c,$$

where $\boldsymbol{D}(\boldsymbol{\sigma}^2)$ gives a diagonal matrix with $\boldsymbol{\sigma}^2$ in its diagonal, $|\boldsymbol{M}|$ is the determinant of a matrix $\boldsymbol{M}$ and $c$ is some constant. We can write the gradients of the log-likelihood function with respect to $\boldsymbol{\mu}$ and $\boldsymbol{\sigma}^2$ as follows:

$$\nabla_{\boldsymbol{\mu}} \log q(\boldsymbol{x}; \boldsymbol{\mu}, \boldsymbol{\sigma}^2) = \boldsymbol{D}(\boldsymbol{\sigma}^2)^{-1}(\boldsymbol{x} - \boldsymbol{\mu}) = (\boldsymbol{x} - \boldsymbol{\mu}) \oslash \boldsymbol{\sigma}^2,$$

$$\nabla_{\boldsymbol{\sigma}^2} \log q(\boldsymbol{x}; \boldsymbol{\mu}, \boldsymbol{\sigma}^2) = \frac{1}{2}\big[(\boldsymbol{x} - \boldsymbol{\mu})^2 \oslash \boldsymbol{\sigma}^2 - \boldsymbol{1}\big] \oslash \boldsymbol{\sigma}^2,$$

where $\boldsymbol{1}$ is an all-ones vector, and $\oslash$ denotes element-wise division. Finally, we write the exact diagonals of the Hessian matrix as

$$\mathrm{diag}(\boldsymbol{H_\mu}) = \mathrm{diag}(\nabla_{\boldsymbol{\mu}}(\boldsymbol{x} - \boldsymbol{\mu}) \oslash \boldsymbol{\sigma}^2) = -\boldsymbol{1} \oslash \boldsymbol{\sigma}^2,$$

$$\mathrm{diag}(\boldsymbol{H_{\sigma^2}}) = \mathrm{diag}\Big(\nabla_{\boldsymbol{\sigma}^2}\big[\frac{1}{2}[(\boldsymbol{x} - \boldsymbol{\mu})^2 \oslash \boldsymbol{\sigma}^2 - \boldsymbol{1}] \oslash \boldsymbol{\sigma}^2\big]\Big) = \big[0.5\boldsymbol{1} - (\boldsymbol{x} - \boldsymbol{\mu})^2 \oslash \boldsymbol{\sigma}^2\big] \oslash \boldsymbol{\sigma}^4.$$

Clearly, the gradient and the exact Hessian diagonals can be computed in $O(n)$. Log-likelihood functions for normal distributions are used in many important problems, such as variational inference and continuous reinforcement learning control.

# B HesScale with Convolutional Neural Networks

Here, we derive the Hessian propagation for convolutional neural networks (CNNs). Consider a CNN with $L-1$ layers followed by a fully connected layer that outputs the predicted output $\boldsymbol{q}$. The CNN filters are parameterized by $\{\boldsymbol{W}_1, ..., \boldsymbol{W}_L\}$, where $\boldsymbol{W}_l$ is the filter matrix at the $l$-th layer with the dimensions $k_{l,1} \times k_{l,2}$, and its element at the $i$th row and the $j$th column is denoted by $W_{l,i,j}$. For the simplicity of this proof, we assume that the number of filters at each layer is one; the proof can be extended easily to the general case. The learning algorithm learns the target function $f^*$ by optimizing the loss $\mathcal{L}$. During learning, the parameters of the neural network are changed to reduce the loss. At the layer $l$, we get the activation output matrix $\boldsymbol{H}_l$ by applying the activation function $\boldsymbol{\sigma}$ to the activation input $\boldsymbol{A}_l$: $\boldsymbol{H}_l = \boldsymbol{\sigma}(\boldsymbol{A}_l)$. We assume here that the activation function is element-wise activation for all layers except for the final layer $L$, where it becomes the softmax function. We simplify notations by defining $\boldsymbol{H}_0 \doteq \boldsymbol{X}$, where $\boldsymbol{X}$ is the input sample. The activation output $\boldsymbol{H}_l$ is then convoluted by the weight matrix $\boldsymbol{W}_{l+1}$ of layer $l+1$ to produce the next activation input: $A_{l+1,i,j} = \sum_{m=0}^{k_{l,1}-1} \sum_{n=0}^{k_{l,2}-1} W_{l+1,m,n} H_{l,(i+m),(j+n)}$. We denote the size of the activation output at the $l$-th layer by $h_l \times w_l$. The backpropagation equations for the described network are given following Rumelhart et al. (1986):

$$
\begin{aligned}
\frac{\partial \mathcal{L}}{\partial A_{l,i,j}} &= \sum_{m=0}^{k_{l+1,1}-1} \sum_{n=0}^{k_{l+1,2}-1} \frac{\partial \mathcal{L}}{\partial A_{l+1,(i-m),(j-n)}} \frac{\partial A_{l+1,(i-m),(j-n)}}{\partial A_{l,i,j}} \\
&= \sum_{m=0}^{k_{l+1,1}-1} \sum_{n=0}^{k_{l+1,2}-1} \frac{\partial \mathcal{L}}{\partial A_{l+1,(i-m),(j-n)}} \sum_{m'=0}^{k_{l+1,1}-1} \sum_{n'=0}^{k_{l+1,2}-1} W_{l+1,m',n'} \frac{\partial H_{l,(i-m+m'),(j-n+n')}}{\partial A_{l,i,j}} \\
&= \sum_{m=0}^{k_{l+1,1}-1} \sum_{n=0}^{k_{l+1,2}-1} \frac{\partial \mathcal{L}}{\partial A_{l+1,(i-m),(j-n)}} W_{l+1,m,n} \sigma'(A_{l,i,j}) \\
&= \sigma'(A_{l,i,j}) \sum_{m=0}^{k_{l+1,1}-1} \sum_{n=0}^{k_{l+1,2}-1} \frac{\partial \mathcal{L}}{\partial A_{l+1,(i-m),(j-n)}} W_{l+1,m,n},
\end{aligned}
\tag{6}
$$

$$
\frac{\partial \mathcal{L}}{\partial W_{l,i,j}} = \sum_{m=0}^{h_l-k_{l,1}} \sum_{n=0}^{w_l-k_{l,2}} \frac{\partial \mathcal{L}}{\partial A_{l,m,n}} \frac{\partial A_{l,m,n}}{\partial W_{l,i,j}} = \sum_{m=0}^{h_l-k_{l,1}} \sum_{n=0}^{w_l-k_{l,2}} \frac{\partial \mathcal{L}}{\partial A_{l,m,n}} H_{l-1,(i+m),(j+n)}.
\tag{7}
$$

In the following, we write the equations for the exact Hessian diagonals with respect to weights $\partial^2 \mathcal{L}/\partial W_{l,i,j}^2$, which requires the calculation of $\partial^2 \mathcal{L}/\partial A_{l,i,j}^2$ first:

$$
\begin{aligned}
\frac{\partial^2 \mathcal{L}}{\partial A_{l,i,j}^2} &= \frac{\partial}{\partial A_{l,i,j}} \left[ \sigma'(A_{l,i,j}) \sum_{m=0}^{k_{l+1,1}-1} \sum_{n=0}^{k_{l+1,2}-1} \frac{\partial \mathcal{L}}{\partial A_{l+1,(i-m),(j-n)}} W_{l+1,m,n} \right] \\
&= \sigma'(A_{l,i,j}) \sum_{m,p=0}^{k_{l+1,2}-1} \sum_{n,q=0}^{k_{l+1,2}-1} \frac{\partial^2 \mathcal{L}}{\partial A_{l+1,(i-m),(j-n)} \partial A_{l+1,(i-p),(j-q)}} \frac{\partial A_{l+1,(i-p),(j-q)}}{\partial A_{l,i,j}} W_{l+1,m,n} \\
&\quad + \sigma''(A_{l,i,j}) \sum_{m=0}^{k_{l+1,2}-1} \sum_{n=0}^{k_{l+1,2}-1} \frac{\partial \mathcal{L}}{\partial A_{l+1,(i-m),(j-n)}} W_{l+1,m,n}
\end{aligned}
$$

$$
\begin{aligned}
\frac{\partial^2 \mathcal{L}}{\partial W_{l,i,j}^2} &= \frac{\partial}{\partial W_{l,i,j}} \left[ \sum_{m=0}^{h_l-k_{l,1}} \sum_{n=0}^{w_l-k_{l,2}} \frac{\partial \mathcal{L}}{\partial A_{l,m,n}} H_{l-1,(i+m),(j+n)} \right] \\
&= \sum_{m,p=0}^{h_l-k_{l,1}} \sum_{n,q=0}^{w_l-k_{l,2}} \frac{\partial^2 \mathcal{L}}{\partial A_{l,m,n} \partial A_{l,p,q}} \frac{\partial A_{l,p,q}}{\partial W_{l,i,j}} H_{l-1,(i+m),(j+n)}
\end{aligned}
$$

Since the calculation of $\partial^2 \mathcal{L}/\partial A_{l,i,j}^2$ and $\partial^2 \mathcal{L}/\partial W_{l,i,j}^2$ depend on the off-diagonal terms, the computation complexity becomes quadratic. Following Becker and Lecun (1989), we approximate the

Hessian diagonals by ignoring the off-diagonal terms, which leads to a backpropagation rule with linear computational complexity for our estimates $\widehat{\frac{\partial^2 \mathcal{L}}{\partial W_{l,i,j}^2}}$ and $\widehat{\frac{\partial^2 \mathcal{L}}{\partial A_{l,i,j}^2}}$:

$$\widehat{\frac{\partial^2 \mathcal{L}}{\partial A_{l,i,j}^2}} \doteq \sigma'(A_{l,i,j})^2 \sum_{m=0}^{k_{l+1,2}-1} \sum_{n=0}^{k_{l+1,2}-1} \widehat{\frac{\partial^2 \mathcal{L}}{\partial A_{l+1,(i-m),(j-n)}^2}} W_{l+1,m,n}^2$$
$$+ \sigma''(A_{l,i,j}) \sum_{m=0}^{k_{l+1,2}-1} \sum_{n=0}^{k_{l+1,2}-1} \frac{\partial \mathcal{L}}{\partial A_{l+1,(i-m),(j-n)}} W_{l+1,m,n}, \tag{8}$$

$$\widehat{\frac{\partial^2 \mathcal{L}}{\partial W_{l,i,j}^2}} \doteq \sum_{m=0}^{h_l-k_{l,1}} \sum_{n=0}^{w_l-k_{l,2}} \widehat{\frac{\partial^2 \mathcal{L}}{\partial A_{l,m,n}^2}} H_{l-1,(i+m),(j+n)}^2. \tag{9}$$

## C  APPROXIMATION QUALITY WITH MNIST DATA

We repeat the experiment shown in Fig. 1 with MNIST data points instead of random data points. The experimental details are the same except for two changes. First, we used a larger network where we changed the number of units in each hidden layer to 32 instead of 16. Second, we performed an optimization update with SGD at each data point. The results shown in Fig. 6 are similar to the results shown in Fig. 1 where HesScale gives a better approximation quality than other methods. This experiment shows that our results hold for realistic settings where learning is involved.

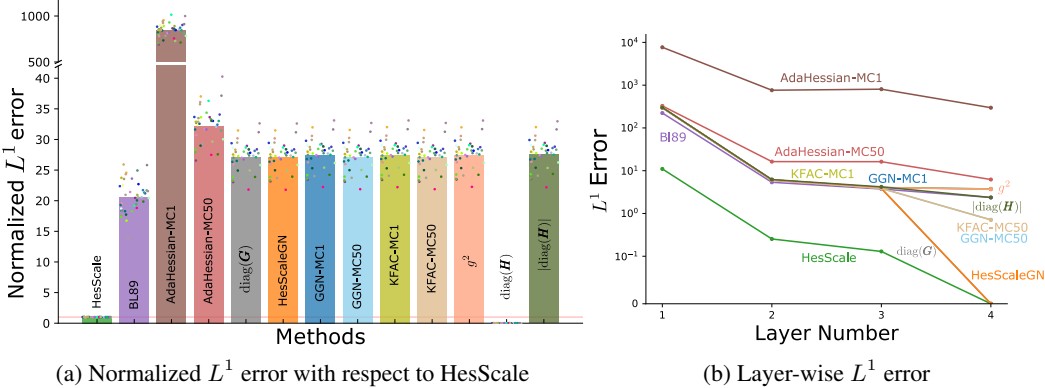

(a) Normalized $L^1$ error with respect to HesScale     (b) Layer-wise $L^1$ error

Figure 6: The averaged error for each method is normalized by the averaged error incurred by HesScale for data points coming from MNIST. We show 40 initialization points with the same colors across all methods. The norm of the vector of Hessian diagonals $|\text{diag}(\boldsymbol{H})|$ is shown as a reference.

# D OPTIMIZATION PLOTS IN THE NUMBER OF EPOCHS

We give the training loss, training accuracy, validation loss, validation accuracy, test loss, and test accuracy for each of the methods we include in our comparison in Fig. 7 and Fig. 8. Moreover, we give the sensitivity plots for $\beta_1$, $\beta_2$, and $\alpha$ for each method in Fig. 9 and Fig. 10.

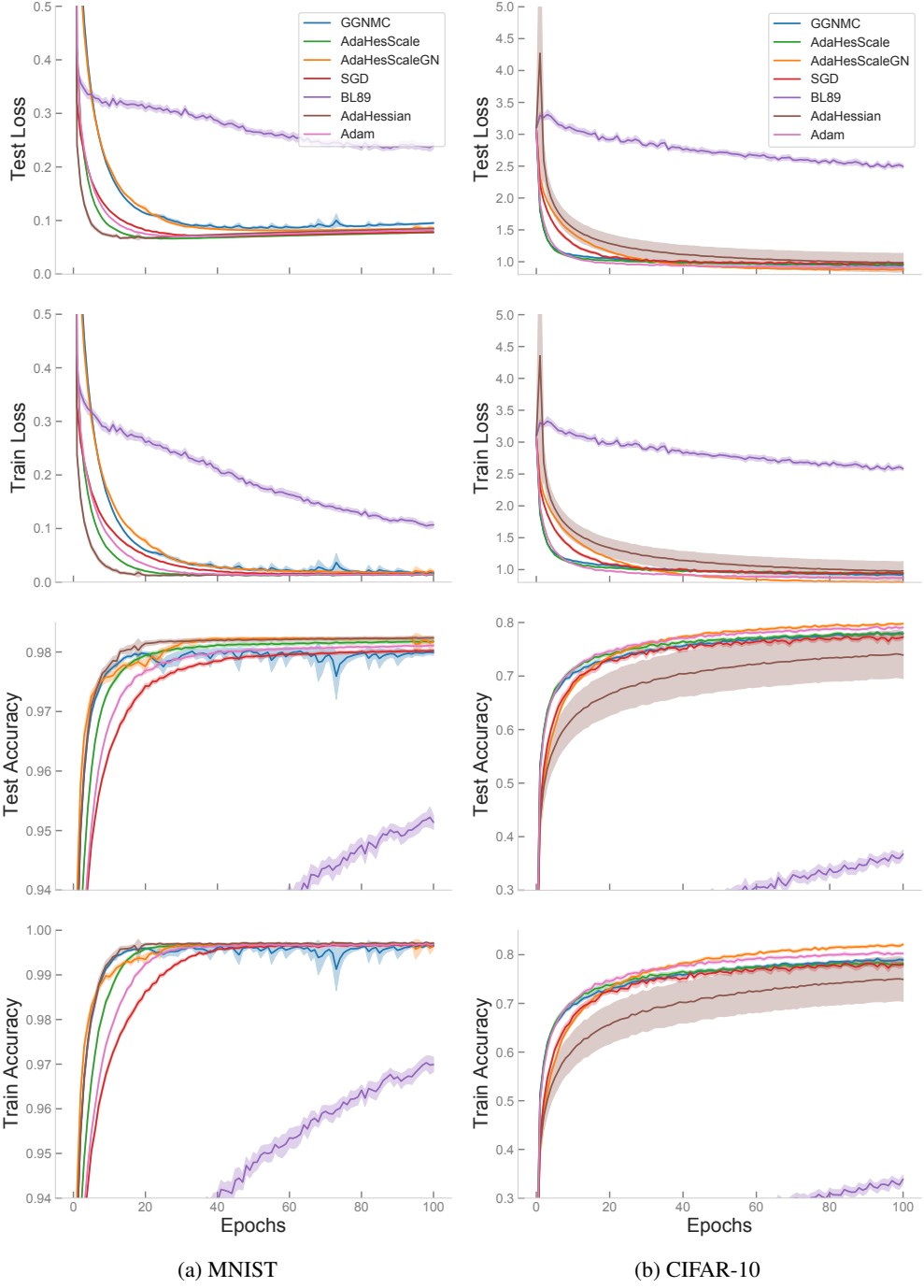

(a) MNIST

(b) CIFAR-10

Figure 7: Learning curves of each algorithm on two tasks, MNIST-MLP and CIFAR-10 3C3D, for 100 epochs. We show the best configuration for each algorithm on the validation set. The best parameter configuration for each algorithm is selected based on the area under the curve for the validation loss.

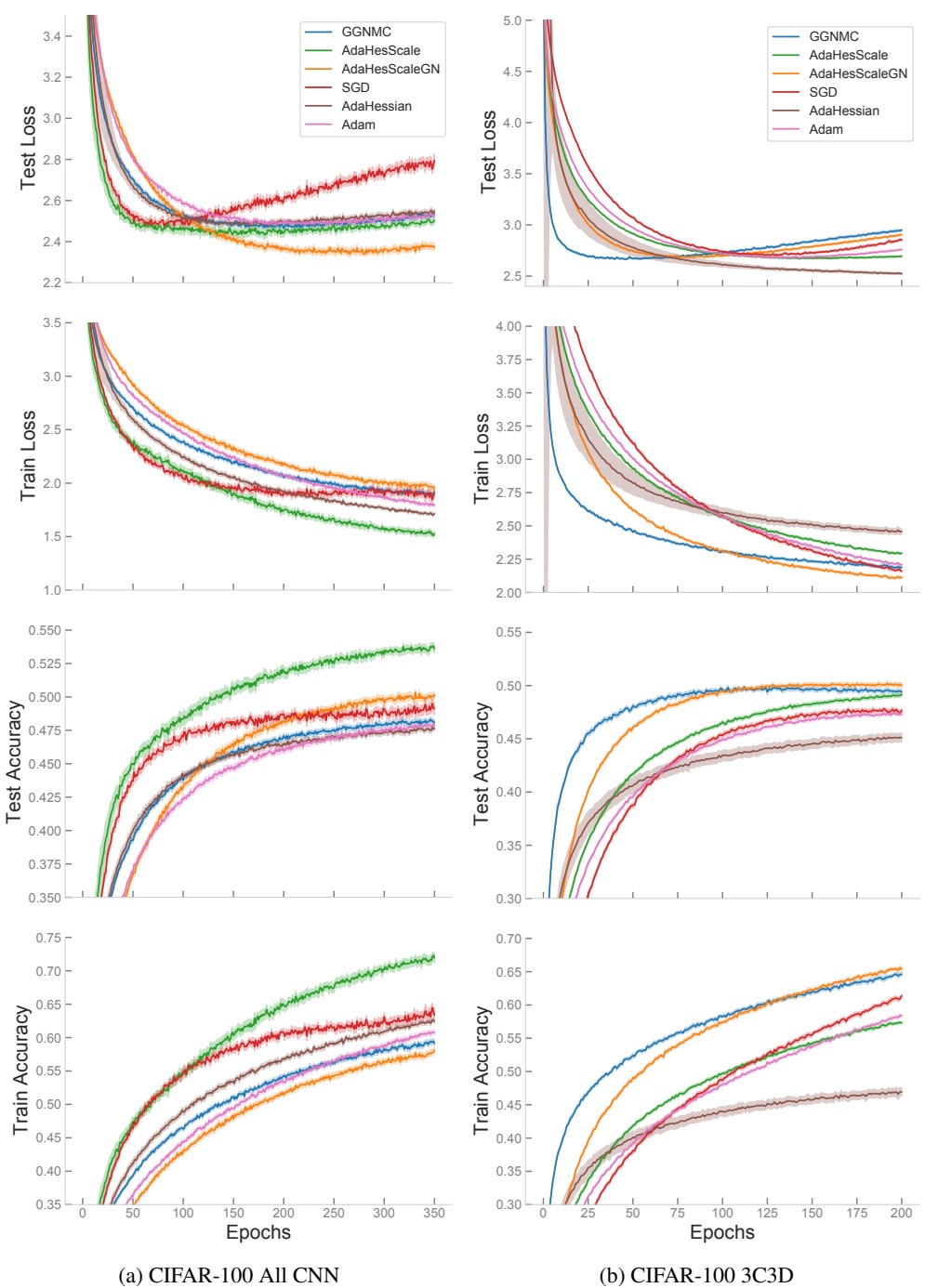

(a) CIFAR-100 All CNN          (b) CIFAR-100 3C3D

Figure 8: Learning Curves of each algorithm on CIFAR-100 with All-CNN and 3C3D architectures, for 100 epochs. We show the best configuration for each algorithm on the validation set. The best parameter configuration for each algorithm is selected based on the area under the curve for the validation loss.

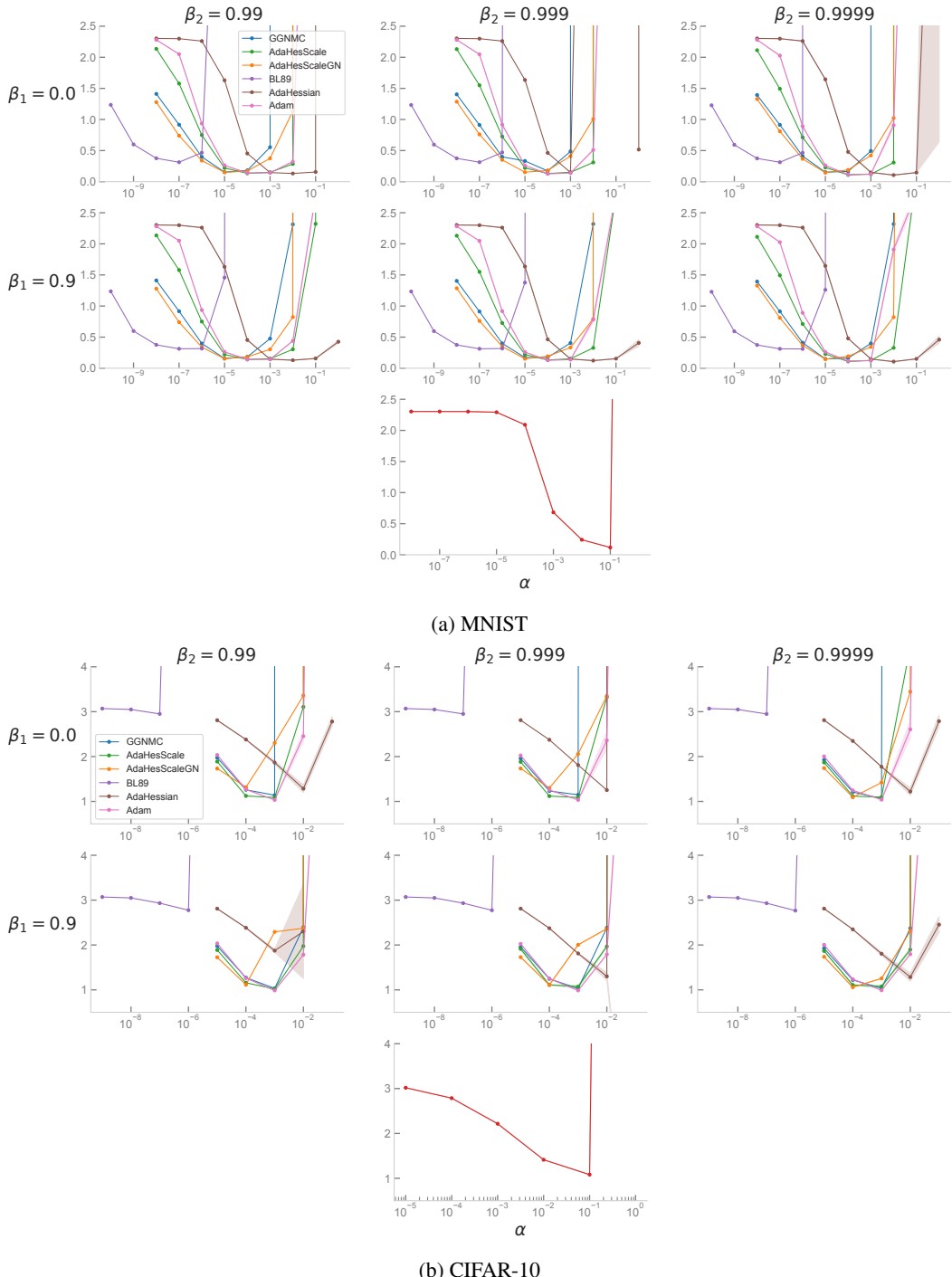

Figure 9: Parameter Sensitivity study for each algorithm on two data sets, MNIST and CIFAR-10. The range of $\beta_2$ is $\{0.99, 0.999, 0.9999\}$ and the range of $\beta_1$ is $\{0.0, 0.9\}$. Each point for each algorithm represents the average test loss given a set of parameters.

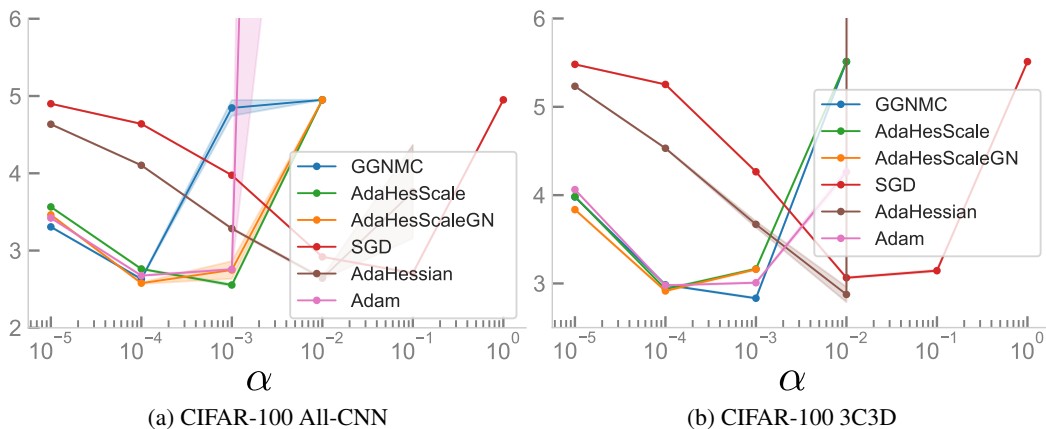

(a) CIFAR-100 All-CNN        (b) CIFAR-100 3C3D

Figure 10: Parameter Sensitivity study for each algorithm on CIFAR-100 with All-CNN and 3C3D architectures. The range of step size is $\{10^{-5}, 10^{-4}, 10^{-3}, 10^{-2}, 10^{-1}, 10^{0}\}$. We choose $\beta_1$ to be equal to 0.9 and $\beta_2$ to be equal to 0.999. Each point for each algorithm represents the average test loss given a set of parameters.

# E  SUMMARY OF OPTIMIZATION RESULTS

We summarize the final performance of AdaHesScale and AdaHesScaleGN against other optimizers on the train sets and test sets, in Table 1 and Table 2 respectively.

Table 1: Performance of optimization methods on the train sets of different problems.

|  | MNIST | CIFAR-10 | CIFAR-100 3C3D | CIFAR-100 ALL-CNN-C |
|---|---|---|---|---|
| AdaHesScale | 99.68% | 78.19% | 57.29% | **71.89**% |
| AdaHesScaleGN | 99.64% | **82.11**% | **65.39**% | 58.10% |
| GGN-MC | 99.64% | 78.92% | 64.62% | 59.14% |
| SGD | 99.65% | 77.93% | 61.37% | 63.37% |
| Adam | 99.67% | 80.24% | 58.36% | 60.82% |
| AdaHessian | **99.71**% | 74.94% | 46.92% | 62.43% |
| BL89 | 96.99% | 33.89% | - | - |

Table 2: Performance of optimization methods on the test sets of different problems.

|  | MNIST | CIFAR-10 | CIFAR-100 3C3D | CIFAR-100 ALL-CNN-C |
|---|---|---|---|---|
| AdaHesScale | 98.17% | 77.98% | 49.13% | **53.59**% |
| AdaHesScaleGN | 98.17% | **79.79**% | **49.99**% | 50.16% |
| GGN-MC | 97.99% | 77.93% | 49.41% | 48.06% |
| SGD | 98.02% | 77.28% | 47.65% | 49.13% |
| Adam | 98.10% | 79.07% | 47.30% | 47.97% |
| AdaHessian | **98.23**% | 73.91% | 45.10% | 47.63% |
| BL89 | 95.13% | 36.74% | - | - |

