# OpenReview forum: "HesScale: Scalable Computation of Hessian Diagonals"
_ICLR.cc/2023/Conference — Submitted to ICLR 2023_

### Official Review · Reviewer_37p6 · 2022-10-21

**Confidence:** 4
**Correctness:** 3
**Technical Novelty And Significance:** 3
**Empirical Novelty And Significance:** 4
**Recommendation:** 8

**Clarity, Quality, Novelty And Reproducibility:**

The paper is clearly written, experimental results are well done (but it'd be great to add examples from other domains).  There is limited  insight or analysis into why the proposed approximation works as well as it does and discussion of where it may work less well.

**Strength And Weaknesses:**

Strength:  A principled and inexpensive approximate second-order optimization method for learning neural nets would be very impactful -- and the paper is taking a shot at this problem.  The method is a small modification of an existing proposal by Becker and Lecun, that observes that the diagonal of the Hessian of the last layer can be computed exactly (rather than approximately) for common loss functions,  but this seems to make a very significant difference in the approximation error (in experiments).  The paper then uses this Hessiagn-diagonal estimator for optimization. Experimental results with image classification show that the method is competitive with other optimizers, and shows the best performance on some if the instances.

Weaknesses:
1) The paper goes into a lot of depth describing the hessian-diagonal approximation, but the actual optimization method is just a paragraph + an algorithm.  Some additional explanation of steps / parameters would make it more readable.

2) I was struck by the impressive results in for HessScale in Figure 1,  but I have some questions on them.  a) Do you have any intuition, insight or analysis on how can the method do so much better than other alternatives (order of 10-20x reduction in error w.r.t best other methods).   Are the discarded off-diagonal terms somehow conveniently cancelling for random problems, or what's going on?  Is the setting of random neural nets on random inputs somehow fortuitous for this evaluation, or does this improvement in Hessian approximation also hold for realistic neural nets in the middle of the training?

3) What limitations are there on when the method can be used for more general neural nets? E.g. can it be applied with dropout, batch-norm layers, or for other architectures (ResNets, RNNs, LSTMs, transformer-style architectures)?  What common losses are not supported?

4) Evaluation.  It would be great to include some examples from other (non-vision) domains and with different achitectures -- e.g. natural language, tabular data, recommendation, maybe speech.  It seems that there are differences in what's the default optimizer in each community, and it could be motivated by artifacts of the structure of the problem / architectures commonly used.

5) For second order methods ideally it would be better to compute the diagonal of the inverse Hessian -- rather than inverting the diagonal of the Hessian? Is that something that can be computed efficiently?  Having access to Hessian-vector products it is possible to estimate the diagonal of the inverse -- using Hutchinson's like methods,  or alternatives, like (this is in context of Gaussian graphical models, but it's essentially an estimator for the diagonal of the inverse of the matrix):
Wilsky, Malioutov, Johnson, "GMRF Variance Approximation using Spliced Wavelet Bases." 2007 IEEE ICASSP, 2007.

6) For AdaHessScaleGCN -- it seems that well into convergence the optimizer occasionally has some large instabilities with large oscillations (e.g. top-left and in Figure 1, and some runs in Fig 6).  Why does this happen, and can it be improved somehow?



**Summary Of The Paper:**

The paper proposes an efficient heuristic approximation to compute the diagonal of the Hessian matrix in neural nets, and empirically shows that it is significantly more accurate than other existing deterministic and stochastic approximations.  The paper then goes on to propose a second-order optimization method that uses these estimated hessian values, and shows on experimental results that the method is competitive and sometimes better than other standard optimizers (SGD, ADAM) in computer vision tasks.

**Summary Of The Review:**

The paper proposed an improved approximate estimator of the diagonal of the Hessian for neural networks, and uses it to propose a second-order method. Experimental results are done well,  but could be more comprehensive.  It'd be very helpful if the authors made an attempt at gaining insight into why the method works as well as it does (and whether there are settings where it works less well) .

---

> ### Author Response · Authors · 2022-11-12
> **Thank you for reviewing our paper (1/2)**
>
> Thank you for your valuable feedback. Please let us know if you have any more questions.
>
> &nbsp;
>
> > I was struck by the impressive results in for HessScale in Figure 1, but I have some questions on them. a) Do you have any intuition, insight or analysis on how can the method do so much better than other alternatives (order of 10-20x reduction in error w.r.t best other methods). Are the discarded off-diagonal terms somehow conveniently canceling for random problems, or what's going on? Is the setting of random neural nets on random inputs somehow fortuitous for this evaluation, or does this improvement in Hessian approximation also hold for realistic neural nets in the middle of the training?
>
> We can provide some intuition why our modification makes much difference. Using the Hessian diagonals propagation equation we give in the paper, we see that the Hessian diagonals at the $l$-th layer are a function of the Hessian diagonals at the $l+1$ layer and the Hessian diagonals at the $l+1$ depend on the elements from the next layer. This means any better approximation at the last layer $L$ will improve the approximation at all layers.
>
> The discarded off-diagonal terms don’t cancel for random inputs. The approximation method we provide is very accurate compared to other methods. For the rebuttal revision paper, we will add a new experiment similar to the one shown in Figure 1 but with MNIST data points (and training)  instead of random inputs with no training. The results of the approximation quality for Hessian diagonals with MNIST show the same pattern. We hope this experiment will increase your confidence about the extension of our results to realistic settings.
>
> &nbsp;
>
> > The paper goes into a lot of depth describing the hessian-diagonal approximation, but the actual optimization method is just a paragraph + an algorithm. Some additional explanation of steps / parameters would make it more readable.
>
> We didn’t go in-depth in describing the choices for our two optimizers, AdaHesScale and AdaHesScaleGGN. This is mainly because we use our approximation with ideas from the Adam optimizer, which we refer to in the paper for interested readers. Similar papers presented their optimizers the same way (e.g., AdaHessian). Since Adam is a well-known optimizer in the community, we omitted detailed descriptions for every step and presented a detailed algorithm instead. We can still add more details if you think they are necessary. Please let us know if you have any other concerns.
>
> &nbsp;
>
> > For second order methods ideally it would be better to compute the diagonal of the inverse Hessian -- rather than inverting the diagonal of the Hessian? Is that something that can be computed efficiently? Having access to Hessian-vector products it is possible to estimate the diagonal of the inverse -- using Hutchinson's like methods, or alternatives, like (this is in context of Gaussian graphical models, but it's essentially an estimator for the diagonal of the inverse of the matrix): Wilsky, Malioutov, Johnson, "GMRF Variance Approximation using Spliced Wavelet Bases." 2007 IEEE ICASSP, 2007.
>
> We agree that utilizing the fact that the Hessian-vector product is cheap can be combined with stochastic approximation (e.g., Hutchinson's estimator), which can result in useful optimizers (e.g., AdaHessian, which we compare to our method). However, we think these kinds of approximation need many samples to achieve good approximation, which might limit their usage. On the other hand, our method is deterministic, so it achieves better approximation than a stochastic approximation, which requires many samples to achieve good approximation (Figure 1).

---

> > ### Author Response · Authors · 2022-11-12
> > **Thank you for reviewing our paper (2/2)**
> >
> >
> > > For AdaHessScaleGGN -- it seems that well into convergence the optimizer occasionally has some large instabilities with large oscillations (e.g. top-left and in Figure 1, and some runs in Fig 6). Why does this happen, and can it be improved somehow?
> >
> > We note here that because we use a scaled-up y-axis, the oscillations seem large. However, it was between 0.25%. As suggested by reviewer HXwV, we will add a table containing the numerical results to avoid any potential confusion.
> >
> > &nbsp;
> >
> > > Evaluation. It would be great to include some examples from other (non-vision) domains and with different achitectures -- e.g. natural language, tabular data, recommendation, maybe speech. It seems that there are differences in what's the default optimizer in each community, and it could be motivated by artifacts of the structure of the problem / architectures commonly used.
> >
> > We agree that although not strictly necessary for this paper, it would be great, and we will consider adding such results if time permits. We also encourage interested readers in the community to use our method in their different applications to see if there is an additional gain compared to first-order methods.
> >
> > &nbsp;
> >
> > > What limitations are there on when the method can be used for more general neural nets? E.g. can it be applied with dropout, batch-norm layers, or for other architectures (ResNets, RNNs, LSTMs, transformer-style architectures)? What common losses are not supported?
> >
> > Our method is applicable in principle to all these cases. However, implementations in some cases would require substantial effort due to the nature of backend autodiff implementations. Although our method is applicable to all these cases, for a certain class of loss, our method will no longer be $O(n)$. This class includes any loss function for which the outer layer’s exact Hessian does not reduce to a diagonal matrix. A specific example is a loss involving a multivariate normal distribution where we care about covariance between variables. Our method will still be applicable to this case by performing exact Hessian diagonal computation just for the outer layer but will suffer $O(n^2)$ computation. Incidentally, even the exact gradient calculation of backpropagation will be $O(n^2)$ in this specific case where we care about covariance, so our method will still not be computationally more complex than backpropagation.
> >
> > &nbsp;
> >
> > We would like to thank you again for reviewing our paper!

---

> > > ### Comment · Reviewer_37p6 · 2022-11-15
> > > **Thanks for the response.**
> > >
> > > Thank you for the detailed reply, I found it helpful.   Overall I'd like to see this paper published -- as it makes a valuable empirical observation that allows to inexpensively compute accurate Hessian-diagonal approximations, which may have important implications for optimization.  Theoretical insight is rather limited as the focus is on experiments, but I'm sure the paper will spur future work trying to understand it better.

---

### Official Review · Reviewer_HXwV · 2022-10-21

**Confidence:** 3
**Correctness:** 3
**Technical Novelty And Significance:** 3
**Empirical Novelty And Significance:** 2
**Recommendation:** 5

**Clarity, Quality, Novelty And Reproducibility:**

Clarity: The paper is clearly written and understandable

Quality: The experiments do not reflect realistic use cases in machine learning which in combination with the insufficient motivation of the method for the machine learning setting makes it hard to believ
e that the method as such is useful in practice. The method is potentially an interesting idea theoretically but the paper does not attemt a theoretical analysis.

Novelty: I am not familiar enough with the literature but the idea seems so simple it would be surprising if it had not been explored elsewhere. Nonetheless, assuming it is novel to try this it is an orig
inal idea to pursue.

Reproducibility: The experiments should be easy enough to reproduce based on the described algorithm details in the paper.

**Strength And Weaknesses:**

Strengths:
- Interesting approach because it utilizes an exact symbolic differentiation approach in a computationally efficient manner
- The authors present convergence results over wall-clock time demonstrating that their approximation is in principle computationally feasible in practice

Weaknesses:
- The background section is potentially a bit extensive on backgroud about block structure that is not necessary for the method proposed in the paper
- It would be interesting to know whether there can be theoretical bounds on the error made from dropping the terms in the symbolic derivative
- Tiny toy example network used for accuracy analysis (3 layers with 16 units, 1 layer with 512 units), not even something like MNIST but just random data examples so it is unclear how the accuracy statements hold in more realistic settings
- Increasing the number of outputs is not a practical application for increasing parameter size since the number of classes does not typically change, furthermore this seems biased as the last layer computation of the Hessian is exact and different from the others. So cost scaling should be done by increasing number of layers and layer width (and much larger input space should be used for the examples)
- The experimental section could be improved by presenting the results in a more structured way also presenting results numerically in tables, the scaling of the plot axes could be misleading
- Not sure if the hyperparam tuning for the different optimizers was sufficient to be convincing
- Need to better motivate the method compared to quasi-Newton type methods for estimating a Hessian diagonal (especially in a full-batch setting)
- Need to motivate the method in the context of nonconvex stochastic estimation or some other application of machine learning. In the end I am not sure sure what the relevance of curvature information is for machine learning. In theory curvature is useful for different contexts, it can be useful to escape saddle point quicker, but that's not how it seems to be used in this paper. The other purpose can be to converge to a an optimum faster when already in a convex region around the optimum. But getting close to an optimum of an empirical loss is not always beneficial for generalization. The authors show test set plots which is unconventional, but maybe they can be viewed like validation set plots. It would have been interesting as well to see how the training set loss behaves to see what the generalization gap is for these applications.

*Questions and remarks*:
- How does this related to edge pushing algorithm (see https://arxiv.org/pdf/2007.15040.pdf)?
- What is the quality of the Hessian approximation when computed on a mini-batch?
- Not sure about beta_1 search space of just two choices, that seems limited?
- Why is the standard error for the CIFAR-10 3C3D AdaHessian run so large?
- Color codes for different optimizers should be identical between plots (Figure 3 a) has AdaHessian in pink and b) has Adam in pink)
- Why does Figure 5 report test loss while Figure 3 and 4 report test accuracy?
- Typo: "sensitivty"


**Summary Of The Paper:**

The authors present a method for approximating the diagonal of the Hessian of neural networks in order to use this approximate curvature information during training.
The approximation proposed by the authors is based on considering the exact symbolic derivative for the Hessian diagonal entries and then dropping terms as necessary during backprop in order to make the computation fast (constant factor of the function evaluation like standard backprop gradient).
The authors analyze the accuracy of their Hessian approximation compared to other baseline methods for approximating curvature information and are able to demonstrate that their approximation has much smaller approximation error in practice for some small toy networks (3 layeers with 16 unites, 1 layer with 512 both with tiny input size).
The authors also present some experimental results demonstrating the usefulness of their Hessian diagonal optimization in terms of reaching high test set accuracy faster.

**Summary Of The Review:**

I do not recommend to accept the paper as the paper does not have either realistic experiments that are meaningful to the machine learning practitioner nor does it present a theoretical analysis of approximation bounds and resulting possible statements about optimizer convergence for the proposed Hessian approximation

---

> ### Author Response · Authors · 2022-11-12
> **Thank you for reviewing our paper (1/2)**
>
> Thank you for your feedback. Please let us know if you have any more questions.
>
> &nbsp;
>
> > The background section is potentially a bit extensive on background about block structure that is not necessary for the method proposed in the paper
>
> In the background section, we gave a quick review of many attempts in the literature to emphasize the potential of diagonal methods. We agree with you that it can be a bit extensive. Would you like will us to move it to the appendix? Would that help accept the paper?
>
> &nbsp;
>
> > It would be interesting to know whether there can be theoretical bounds on the error made from dropping the terms in the symbolic derivative
>
> We are interested in finding the approximation bound of our method compared to the exact diagonals. The focus of this paper is empirical, so we left the theoretical analyses to future works. Do you think they can be left out, and the current paper would still be a considerable contribution to the community?
>
> Some other theoretical analyses, such as convergence proof, are already shown in the literature for methods that use Hessian diagonals. We will add a sentence in our rebuttal revision referring the reader to the convergence proof by [Yao et al. (2021)](https://arxiv.org/pdf/2006.00719.pdf)).
>
> &nbsp;
>
> > Tiny toy example network used for accuracy analysis (3 layers with 16 units, 1 layer with 512 units), not even something like MNIST but just random data examples so it is unclear how the accuracy statements hold in more realistic settings
>
> We will add a new experiment using MNIST in the appendix for our rebuttal revision. Using larger networks is computationally very expensive since we have to compute the exact Hessian for comparisons; therefore, we compare using a small network. The results of the approximation quality for Hessian diagonals with MNIST show the same pattern we have in Figure 1. We hope this experiment will increase your confidence about the extension of our results to realistic settings. Please let us know if you have any questions.
>
> &nbsp;
>
> > Increasing the number of outputs is not a practical application for increasing parameter size since the number of classes does not typically change, furthermore this seems biased as the last layer computation of the Hessian is exact and different from the others. So cost scaling should be done by increasing number of layers and layer width (and much larger input space should be used for the examples)
>
> The number of classes doesn’t change, but they vary from one dataset/task to another. The computational cost of an optimizer with a network of 10 classes is different when the number of classes is 1000, depending on its scalability. Each point in our figure represents a different problem, not a single changing problem.
> We also presented the cost scaling by increasing the number of layers, which is shown in Fig. 2b. Our results are unbiased since Fig. 3 is a computational time comparison. There is no computational gain from using the exact diagonals compared to other methods with linear computational complexity, such as GGN-MC, which uses an estimator for the diagonals. Please let us know if you have any other concerns after our clarification.
>
> &nbsp;
>
> > The experimental section could be improved by presenting the results in a more structured way also presenting results numerically in tables, the scaling of the plot axes could be misleading
>
> We will update our paper to make it more structured as much as possible, given the page limit. In addition, we will add a table showing the numerical results. Thank you for this suggestion to improve the paper’s readability.
>
> &nbsp;
>
> > Not sure if the hyperparam tuning for the different optimizers was sufficient to be convincing
>
> Can you please explain why? We would like to have convincing hyperparameter tuning, so we did a hyperparameter search for the step size, $\beta_1$, and $\beta_2$. The detailed sensitivity plots and train/test learning curves are shown in Appendix C. We couldn’t fit these many figures in the main body. Please check and let us know if you have any other concerns.
>
> &nbsp;
>
> > Need to better motivate the method compared to quasi-Newton-type methods for estimating a Hessian diagonal (especially in a full-batch setting)
>
> To our knowledge, we don’t know any quasi-newton method that approximates the inverse of the Hessian diagonals efficiently. Can you please refer to some quasi-newton methods with linear computational complexity? We would like to add them to our introduction and background sections.

---

> > ### Author Response · Authors · 2022-11-12
> > **Thank you for reviewing our paper (2/2)**
> >
> > > The authors show test set plots which is unconventional, but maybe they can be viewed like validation set plots. It would have been interesting as well to see how the training set loss behaves to see what the generalization gap is for these applications.
> >
> > We show the performance (loss and accuracy plots) on the training/test sets in Appendix C. What we show in the paper body is only the test accuracy with the number of epochs and with time. The rest of the plots exist in the appendix. Please check and let us know if you have any questions.
> >
> > &nbsp;
> >
> > > Need to motivate the method in the context of nonconvex stochastic estimation or some other application of machine learning. In the end I am not sure what the relevance of curvature information is for machine learning. In theory curvature is useful for different contexts, it can be useful to escape saddle point quicker, but that's not how it seems to be used in this paper. The other purpose can be to converge to a an optimum faster when already in a convex region around the optimum. But getting close to an optimum of an empirical loss is not always beneficial for generalization
> >
> > We follow the direction of previous works (e.g., AdaHessian and K-FAC), which resulted in a fruitful direction and showed that curvature information appears to be useful for machine learning. A full investigation is yet to be made for why they perform better than first-order methods since there are many confounding factors. In our broad impact section, we show how our method can be potentially useful for improving the works on neural network pruning and the works addressing catastrophic forgetting.
> >
> > &nbsp;
> >
> > **Response to the questions and remarks:**
> > - We are unfamiliar with the work of the edge-pushing algorithm, but it appears that this algorithm focuses on the whole Hessian, not the diagonals. We will read the paper more carefully and respond soon.
> > - Using mini-batches for comparing the Hessian diagonals approximation wouldn’t create a fair comparison since all methods we compare, approximate the example-by-example Hessian diagonals. The mini-batch average approximate Hessian diagonals against the mini-batch average exact diagonals wouldn’t be an indicator of the approximation quality. However, we tried with minim-batches after reading your comment, and the results show the same pattern where HesScale outperforms other methods.
> > - The search space of $\beta_1$ is limited because of the computational constraint. We follow the same pattern in other papers (e.g., Adam), which uses the same search space we use.
> > - The standard error is large because AdaHessian is a stochastic approximation method, and we use a single Monte Carlo sample (which is what is used in the original paper). The approximation quality is not high, as we show in Fig. 1.
> > - The color inconsistency will be fixed in the updated paper. Thank you for pointing it out.
> > - Figure 5 reports the test loss, not accuracy, since we select the best set of hyperparameters based on the loss, not accuracy. We show the rest of the plots in the appendix.
> > - The typo for the word sensitivity will be fixed in the updated paper.
> >
> > &nbsp;
> >
> > > I do not recommend to accept the paper as the paper does not have either realistic experiments that are meaningful to the machine learning practitioner nor does it present a theoretical analysis of approximation bounds and resulting possible statements about optimizer convergence for the proposed Hessian approximation
> >
> > We believe that our response has addressed most of the issues (including the convergence proof) and we will upload the updated paper soon addressing the rest of the issues. The only remaining issue is the approximation bounds which we left for future work. We hope that you can reconsider your score if you think the issues are/will be addressed.
> >
> > &nbsp;
> >
> > We would like to thank you again for reviewing our paper!

---

> > > ### Comment · Reviewer_HXwV · 2022-11-16
> > > **Thank you for your response**
> > >
> > > > In the background section, we gave a quick review of many attempts in the literature to emphasize the potential of diagonal methods. We agree with you that it can be a bit extensive. Would you like will us to move it to the appendix? Would that help accept the paper?
> > >
> > > The comment about extensive background was meant to help improve balance and readability. It does not impact much whether I think the paper meets the bar.
> > >
> > > > We are interested in finding the approximation bound of our method compared to the exact diagonals. The focus of this paper is empirical, so we left the theoretical analyses to future works. Do you think they can be left out, and the current paper would still be a considerable contribution to the community?
> > >
> > > I agree that empirical work can be a relevant contribution. But for purely empirical work I would appreciate if the paper considered relevant state of the art model architectures from various domains.
> > >
> > > > To our knowledge, we don’t know any quasi-newton method that approximates the inverse of the Hessian diagonals efficiently. Can you please refer to some quasi-newton methods with linear computational complexity?
> > >
> > > For example this paper: https://arxiv.org/abs/2009.13586
> > >
> > > Thanks for extensively addressing my review. I will change my score to marginally below acceptance threshold. I think the method is interesting and should be published. I just feel like the paper needs more work to demonstrate practical use if there is no theoretical analysis.

---

### Official Review · Reviewer_hQqe · 2022-10-23

**Confidence:** 4
**Correctness:** 2
**Technical Novelty And Significance:** 2
**Empirical Novelty And Significance:** 1
**Recommendation:** 3

**Clarity, Quality, Novelty And Reproducibility:**

The motivation for the study, the proposed algorithm, and the details of the experiments are clearly described. However, I believe the comparison method with existing methods (errors, computation time, training speed) is not appropriate.


**Strength And Weaknesses:**

Strengths
- The computational cost, training convergence, and sensitivity to hyperparameters of several diagonal second-order optimization and adaptive gradient methods are compared. (The validity of the comparison is questionable, though.)

Weaknesses
- The proposed HesScale appears to be a very crude approximation without justification.
    - HesScale ignores the off-diagonal elements of Hessian w.r.t. activation, but no theoretical or intuitive justification for this has been provided. Although the experimental results shown (Figure 1) indicate that HesScale has a relatively small error, it is unclear whether this is true in other tasks.
    - The recursive computation of the Hessian is already discussed by Botev et al. (2017) (https://arxiv.org/pdf/1706.03662.pdf) and Dangel et al. (2020) (https://openreview.net/forum?id=BJlrF24twB), where they “back-propagate” the Hessian w.r.t. activation including the off-diagonal elements.
- The validity of the comparison is questionable.
    - It is unfair to compare only the diagonal elements of K-FAC in Figure 1. K-FAC approximates the layer-wise Fisher matrix, including off-diagonal elements, by sacrificing the accuracy of the diagonal elements. Therefore, the error of the layer-wise Fisher approximation should be compared. If only the diagonal elements matter, then a computational approach that only calculates the diagonal elements of the K-FAC matrix should be used to compare the computing time.
    - Although diag(H) is used as the grand truth, H (Hessian) is not necessarily a valuable measure of information for second-order optimization in deep learning because the generalized Gauss-Newton and Fisher matrix is preferred in nonconvex optimization (e.g., https://www.cs.toronto.edu/~jmartens/docs/Deep_HessianFree.pdf).
    - Similarly, the error with diag(H) is not a good indicator because what second-order optimization uses is the inverse of the (damped) second-order information: H_inv = (H + dI)^{-1}, where d > 0 is the damping value corresponding to \epsilon in Algorithm 2. Depending on the choice of d (it could be 1e-1 to 1e-4 in practice https://arxiv.org/pdf/1806.03884.pdf), the importance of the accuracy of estimating diag(H) will be limited.
    - Based on the above, I suggest reviewing the comparison approaches.
- The advantage of AdaHesScale over Adam is unclear.
    - AdaHesScale (Algorithm 2) uses the 'square root of' the inverse of the approximate diagonal Hessian as the preconditioner of the gradient, which does not correspond to the Newton-Raphson method. While the motivation for this study comes from improving approximation accuracy (and computational speed) of the Newton-Raphson method in large settings, it is unclear how much of the second-order information is utilized in AdaHesScale since it takes an update rule almost identical to that of Adam.
    - According to the leaderboard for the CIFAR-100 classification task with All-CNN-C in DeepOBS (https://deepobs.github.io/leaderboardP6.html), Adam achieves a test accuracy of >61% at about 350 epochs, which is much better than Adam (47%) and AdaHesScale (53%) in Figure 4(b). Furthermore, training with momentum SGD achieves >66%. Even if it is difficult to reproduce the leaderboard results perfectly, I do not believe it is reasonable to use a result about 12% lower than the existing results as a baseline for Adam.

**Summary Of The Paper:**

This study proposes HesScale, which can quickly compute the diagonal elements of loss Hessian in deep neural networks by back-propagating only the diagonal component of Hessian w.r.t. activation. For an MLP, it is shown that HesScale estimates a more accurate diagonal Hessian at a lower computational cost than existing methods such as AdaHessian. This study also proposes AdaHesScale, which incorporates the approximate diagonal Hessian calculated by HesScale into an Adam-style update rule. The experiment results show that AdaHesScale (and its variant) can achieve a certain test accuracy faster than SGD and Adam in training CNNs and MLPs for image classification tasks.


**Summary Of The Review:**

While the computation of fast and accurate second-order information is indeed an important topic, questions remain about the validity of the approximations in the proposed HesScale and AdaHesScale, their superiority over existing methods, and the generality of the proposed method’s effectiveness. The quality of this study could be improved by more profound validation of the approximations used in HesScale (ignoring non-diagonal elements of the back-propagated Hessian w.r.t. activation) and more appropriate comparisons with existing second-order methods.

---

> ### Author Response · Authors · 2022-11-12
> **Thank you for reviewing our paper (1/2)**
>
> Thank you for your valuable feedback. Please let us know if you have any more questions.
>
> &nbsp;
>
> > The proposed HesScale appears to be a very crude approximation without justification. HesScale ignores the off-diagonal elements of Hessian w.r.t. activation, but no theoretical or intuitive justification for this has been provided.
>
> The reason why we dropped the off-diagonal elements is that we are interested in achieving scalable second-order methods with linear time computational complexity. Computing the Hessian diagonals requires quadratic computation, which results in methods that are not scalable. For example, methods that backpropagate the full Hessian matrices are out of the scope of this paper since their computations are very expensive. The justification is mentioned in the fifth paragraph of section 3, which reads as follows: "Since, the calculation of $\partial^2 \mathcal{L}/{\partial a^2_{l,i}}$ depends on the off-diagonal terms, the computation complexity becomes quadratic. Following Becker and Lecun (1989), we approximate the Hessian diagonals by ignoring the off-diagonal terms, which leads to a backpropagation rule with linear computational complexity for our estimates."
>
> &nbsp;
>
> > The recursive computation of the Hessian is already discussed by Botev et al. (2017) (https://arxiv.org/pdf/1706.03662.pdf) and Dangel et al. (2020) (https://openreview.net/forum?id=BJlrF24twB), where they “back-propagate” the Hessian w.r.t. activation including the off-diagonal elements.
>
> Hessian diagonals propagation is discussed before in the literature which is cited in the paper (e.g., [Becker & LeCun 1989](http://yann.lecun.com/exdb/publis/pdf/becker-lecun-89.pdf)). Later works introduced variations for full Hessian matrices propagation (e.g., [Mizutani & Dreyfus 2008](https://www.sciencedirect.com/science/article/abs/pii/S0893608007002729), [Botev et al. 2017](https://arxiv.org/pdf/1706.03662.pdf), [Dangel et al. 2020](https://openreview.net/forum?id=BJlrF24twB)). However, we focused on scalable methods, and that’s why we omitted discussing these variants. After reading your review, we now think it’s important to make such a contrast between scalable and non-scalable Hessian propagation methods. We will add a sentence clarifying the difference between the methods mentioned above.
>
> &nbsp;
>
> > Although the experimental results shown (Figure 1) indicate that HesScale has a relatively small error, it is unclear whether this is true in other tasks.
>
> We will add a new experiment using MNIST in the appendix for our rebuttal revision. The results of the approximation quality for Hessian diagonals with MNIST show the same pattern we have in Figure 1. We hope this experiment will increase your confidence about the extension of our results to other tasks. Please let us know if you have any questions.
>
> &nbsp;
>
> > The advantage of AdaHesScale over Adam is unclear. AdaHesScale (Algorithm 2) uses the 'square root of' the inverse of the approximate diagonal Hessian as the preconditioner of the gradient, which does not correspond to the Newton-Raphson method. While the motivation for this study comes from improving approximation accuracy (and computational speed) of the Newton-Raphson method in large settings, it is unclear how much of the second-order information is utilized in AdaHesScale since it takes an update rule almost identical to that of Adam.
>
> Thank you for pointing out this. It’s a typo. AdaHesScale uses the inverse of the approximate diagonal Hessian, not the square root of it. We are squaring and then taking the square root, hence keeping the units correct. This typo will be fixed in our rebuttal revision paper. Please let us know if you have any other concerns.
> The difference between Adam and AdaHesScale is in the use of approximate Hessian diagonals instead of the gradient information. Using HesScale, one can come up with many variants. We chose to make our method similar to existing algorithms, such as Adam and AdaHessian, to have a fair comparison. The main difference between the compared optimizers is their preconditioner of the gradient, which is what we compare.

---

> > ### Author Response · Authors · 2022-11-12
> > **Thank you for reviewing our paper (2/2)**
> >
> > > According to the leaderboard for the CIFAR-100 classification task with All-CNN-C in DeepOBS (https://deepobs.github.io/leaderboardP6.html), Adam achieves a test accuracy of >61% at about 350 epochs, which is much better than Adam (47%) and AdaHesScale (53%) in Figure 4(b). Furthermore, training with momentum SGD achieves >66%. Even if it is difficult to reproduce the leaderboard results perfectly, I do not believe it is reasonable to use a result about 12% lower than the existing results as a baseline for Adam.
> >
> > Thank you for pointing out this comparison. We used the benchmarking task of All-CNN-C with ELU activations instead of ReLU, which is reported on their website. We mentioned in the paper that we made this decision to be able to differentiate between the performance of AdaHesScale and AdaHesScaleGGN. This is why there is a difference between the results we show and the results reported by DeepOBS.
> > We will write a clearer sentence that we use a variation of DeepOBS tasks since we did not use ReLU activations. Thank you for letting us know that this could have potentially caused confusion for future readers. If you would like to see the results for ReLUs to make sure they are close to the leaderboard results on DeepOBS, please let us know to run these experiments.
> >
> > &nbsp;
> >
> > > The validity of the comparison is questionable. It is unfair to compare only the diagonal elements of K-FAC in Figure 1. K-FAC approximates the layer-wise Fisher matrix, including off-diagonal elements, by sacrificing the accuracy of the diagonal elements. Therefore, the error of the layer-wise Fisher approximation should be compared. If only the diagonal elements matter, then a computational approach that only calculates the diagonal elements of the K-FAC matrix should be used to compare the computing time.
> >
> > Since we cannot isolate the computation for diagonal elements, we have to compute the whole matrix and extract the diagonals. We are unaware of any computational approach that only calculates the diagonals for K-FAC. For that reason, do you think that K-FAC should be removed from the comparison?
> >
> > &nbsp;
> >
> > > Although diag(H) is used as the grand truth, H (Hessian) is not necessarily a valuable measure of information for second-order optimization in deep learning because the generalized Gauss-Newton and Fisher matrix is preferred in nonconvex optimization (e.g., https://www.cs.toronto.edu/~jmartens/docs/Deep_HessianFree.pdf).
> >
> > Our method, HesScale, provides a better approximation for Hessian diagonals in neural networks. The applications that can benefit from better approximation go beyond optimization, as we discussed in the broader impact section.
> > We agree that a better approximation for Hessian diagonals is not necessarily better for optimization, and approximating the generalized gauss newton (GGN) matrix is preferred. Therefore, we introduced an approximation for GGN diagonals which we referred to as HesScaleGGN in our paper.
> >
> > &nbsp;
> >
> > > Similarly, the error with diag(H) is not a good indicator because what second-order optimization uses is the inverse of the (damped) second-order information: H_inv = (H + dI)^{-1}, where d > 0 is the damping value corresponding to \epsilon in Algorithm 2. Depending on the choice of d (it could be 1e-1 to 1e-4 in practice https://arxiv.org/pdf/1806.03884.pdf), the importance of the accuracy of estimating diag(H) will be limited.
> >
> > We agree that the use of damped second-order information with high damping value might depend less on the accuracy of estimating the Hessian diagonals. However, with smaller values that are widely used (e.g., 1e-8), the importance of accuracy is large. Our experiments show that our optimizers are outperforming Adam with a damping of 1e-8 due to the better approximation.
> >
> > &nbsp;
> >
> > > Correctness: 2: Several of the paper’s claims are incorrect or not well-supported.
> >
> > We noticed that you gave a score of two for correctness which means that you think there are several of the paper’s claims are incorrect or not well-supported. Can you please let us know what claims?
> >
> > &nbsp;
> >
> > We believe that our response has addressed most of the issues and we will upload the updated paper soon addressing the rest of the issues. We hope that you can reconsider your score if you think the issues are/will be addressed.
> >
> > We would like to thank you again for reviewing our paper!

---

> > > ### Comment · Reviewer_hQqe · 2022-11-15
> > > **Thank you for reply (1/2)**
> > >
> > >
> > >
> > > > The reason why we dropped the off-diagonal elements is that we are interested in achieving scalable second-order methods with linear time computational complexity.
> > >
> > > I don't think this justifies ignoring the off-diagonal elements of Hessian w.r.t. activations when calculating the diagonal elements of Hessian w.r.t. parameters accurately. I believe that more compelling insight and analysis into what factors determine the diagonal elements of Hessian w.r.t. parameters and when and why the off-diagonal elements of Hessian w.r.t. activations can be ignored are needed for justification.
> > >
> > >
> > > > we now think it’s important to make such a contrast between scalable and non-scalable Hessian propagation methods. We will add a sentence clarifying the difference between the methods mentioned above.
> > >
> > > I agree with this point. However, the proposed method is faster than existing methods simply because it ignores most of the elements of Hessian w.r.t. activation (the non-diagonal elements) for calculating the diagonal Hessian w.r.t. parameters. Unless there is a justification for the approximation, a “scalable" method is not always reliable.
> > >
> > >
> > > > We will add a new experiment using MNIST in the appendix for our rebuttal revision.
> > >
> > > Additional results are helpful, but I do not believe they are critical to justify the approximation.
> > >
> > >
> > > > We are squaring and then taking the square root, hence keeping the units correct. This typo will be fixed in our rebuttal revision paper.
> > >
> > > Thank you for addressing this. Yet, the motivation behind “squaring and then taking the square root” is unclear. Why don’t you simply take the moving average of the diagonal Hessians?

---

> > > > ### Comment · Reviewer_hQqe · 2022-11-15
> > > > **Thank you for reply (2/2)**
> > > >
> > > > >  (my comment) According to the leaderboard for the CIFAR-100 classification task with All-CNN-C in DeepOBS (https://deepobs.github.io/leaderboardP6.html), Adam achieves a test accuracy of >61% at about 350 epochs, which is much better than Adam (47%) and AdaHesScale (53%) in Figure 4(b). Furthermore, training with momentum SGD achieves >66%. Even if it is difficult to reproduce the leaderboard results perfectly, I do not believe it is reasonable to use a result about 12% lower than the existing results as a baseline for Adam.
> > > >
> > > > > We used the benchmarking task of All-CNN-C with ELU activations instead of ReLU, which is reported on their website. (…) If you would like to see the results for ReLUs to make sure they are close to the leaderboard results on DeepOBS, please let us know to run these experiments.
> > > >
> > > > Yes, training results with ReLUs are helpful. Still, I do not believe a training result with a 12% lower accuracy than the baseline is reasonable data to compare the new optimizer, even if another activation function (ELU) is used.
> > > >
> > > >
> > > > > we cannot isolate the computation for diagonal elements, we have to compute the whole matrix and extract the diagonals.
> > > >
> > > > This is not true. As $diag(A\otimes B)=diag(A)\otimes diag(B)$ and $diag(A)=diag(E[aa^T])=E[a\odot a]$, one can get the diagonal elements of a K-FAC matrix in a cheaper cost. Yet, as I mentioned, K-FAC calculates the (approximate) full layer-wise FIM by sacrificing the accuracy of the diagonal elements. Therefore, it is fair to compare the approximation quality of the layer-wise FIM (K-FAC should be more costly but more accurate than a diagonal approximation — there is a time-accuracy trade-off.)
> > > >
> > > >
> > > > > Our method, HesScale, provides a better approximation for Hessian diagonals in neural networks.
> > > >
> > > > Again, this statement requires a justification on why/when off-diagonal elements of Hessian w.r.t activation can be ignored.
> > > >
> > > >
> > > > > with smaller values that are widely used (e.g., 1e-8), the importance of accuracy is large
> > > >
> > > > I don’t think such a small damping value is widely used for optimization with FIM or GGN (e.g., https://arxiv.org/pdf/1806.03884.pdf, https://arxiv.org/pdf/2107.01739.pdf, https://ieeexplore.ieee.org/document/9123671).
> > > >
> > > > > Our experiments show that our optimizers are outperforming Adam with a damping of 1e-8 due to the better approximation.
> > > >
> > > > As I pointed out, I do not think the training results are reasonable for the optimizer comparison. As we discussed, the diagonal Hessian is not necessarily a good indicator of the quality of the preconditioner. Adam’s goal is to minimize the regret in online learning, while HesScale’s goal is to estimate the diagonal elements of Hessian. Hence, it is hard to say that HesScale is “the better approximation” for training neural networks.
> > > >
> > > >
> > > >
> > > > > We noticed that you gave a score of two for correctness which means that you think there are several of the paper’s claims are incorrect or not well-supported. Can you please let us know what claims?
> > > >
> > > >
> > > > As I summarized in the weaknesses, (i) the validity of HesScale’s approximation (ignoring the off-diagonal elements of Hessian w.r.t. activations) is not well-supported, (ii) the statement that “HesScale is the better approximation” is not well-supported because the comparison is not fair or the criterion is not appropriate, and (iii) the superiority of AdaHesScale over the other optimizers is not well-supported because the training results are with 12% lower accuracy than the existing baseline results.
> > > >
> > > >
> > > > I appreciate the detailed replies by the authors, but I believe they do not address my concerns.

---

> > > > > ### Author Response · Authors · 2022-11-19
> > > > > **Response to reviewer hQqe (1/3)**
> > > > >
> > > > > Thank you for your detailed reply! Here is our response:
> > > > >
> > > > > > (ours) We will add a new experiment using MNIST in the appendix for our rebuttal revision.
> > > > >
> > > > > > (reviewer) Additional results are helpful, but I do not believe they are critical to justify the approximation.
> > > > >
> > > > > We would like to notify the reviewer that we added this new experiment in response to the reviewer’s other comment, not as a justification for approximation. More specifically, the previous concern was as follows:
> > > > > > (reviewer) Although the experimental results shown (Figure 1) indicate that HesScale has a relatively small error, it is unclear whether this is true in other tasks.
> > > > >
> > > > > Hence, we would request the reviewer to let us address the concerns separately. If the reviewer allows us to address the concerns on a point-by-point basis, does the reviewer agree that this result addresses the other concern? If not, can the reviewer specify how else that concern can be addressed to their satisfaction?
> > > > >
> > > > > &nbsp;
> > > > >
> > > > > > According to the leaderboard for the CIFAR-100 classification task with All-CNN-C in DeepOBS (https://deepobs.github.io/leaderboardP6.html), Adam achieves a test accuracy of >61% at about 350 epochs, which is much better than Adam (47%) and AdaHesScale (53%) in Figure 4(b). Furthermore, training with momentum SGD achieves >66%. Even if it is difficult to reproduce the leaderboard results perfectly, I do not believe it is reasonable to use a result about 12% lower than the existing results as a baseline for Adam.
> > > > >
> > > > > > Still, I do not believe a training result with a 12% lower accuracy than the baseline is reasonable data to compare the new optimizer, even if another activation function (ELU) is used.
> > > > >
> > > > > Here, we want to respectfully clarify that the 12% is imprecise. We think the reviewer compared DeepOBS on the train set with our results on the test set. The numbers are different when comparing the results from the same category. Here is a table summarizing the comparison.
> > > > >
> > > > > |  | Train (ours) | Train (DeepOBS) | Test (ours) | Test(DeepOBS) |
> > > > > |---|---|---|---|---|
> > > > > | SGD  |63.37% | 69.31% | 49.13% | 57.06% |
> > > > > | Adam  | 60.82%  | 61.52%  | 47.97% | 56.15%  |
> > > > >
> > > > > We hope that this clarification will increase your confidence in our results.
> > > > >
> > > > > &nbsp;
> > > > >
> > > > > > (ours) we cannot isolate the computation for diagonal elements, we have to compute the whole matrix and extract the diagonals.
> > > > >
> > > > > > (reviewer) This is not true. As diag(A⊗B)=diag(A)⊗diag(B) and diag(A)=diag(E[aaT])=E[a⊙a], one can get the diagonal elements of a K-FAC matrix in a cheaper cost. Yet, as I mentioned, K-FAC calculates the (approximate) full layer-wise FIM by sacrificing the accuracy of the diagonal elements. Therefore, it is fair to compare the approximation quality of the layer-wise FIM (K-FAC should be more costly but more accurate than a diagonal approximation — there is a time-accuracy trade-off.)
> > > > >
> > > > > We agree with the reviewer, and we take our statement about the inability to isolate the diagonal elements in K-FAC back. What is suggested by the reviewer is what the method of GGNMC does, which we compare our method against. After this discussion, we believe that K-FAC should be removed from the comparison.  We hope this will increase confidence in our comparison and make it more fair. Thank you for your help improve the quality of the paper!
> > > > >
> > > > > &nbsp;
> > > > >
> > > > > > (ours) Our method, HesScale, provides a better approximation for Hessian diagonals in neural networks.
> > > > >
> > > > > > (reviewer) Again, this statement requires a justification on why/when off-diagonal elements of Hessian w.r.t activation can be ignored.
> > > > >
> > > > > Ignoring off-diagonal terms can indeed hurt. That’s why the Becker and LeCun 1989 paper did not become popular, although it was developed with scalability in mind. Our results suggest that ignoring the off-diagonal terms hurt more for the earlier layers. And this is precisely why we don’t ignore the off-diagonal element in the last layer, which increases the approximation for all earlier layers too. Hence, our work is a small advancement in the line of work for scalable methods of computing scalable diagonals.
> > > > >
> > > > > &nbsp;
> > > > >
> > > > > > I believe that more compelling insight and analysis into what factors determine the diagonal elements of Hessian w.r.t. parameters and when and why the off-diagonal elements of Hessian w.r.t. activations can be ignored are needed for justification.
> > > > >
> > > > > When we can only about the quality of the Hessian diagonals, and we don’t have any computational constraints, we should use the exact diagonals, which would require $O(n^2)$. However, this paper is focused on the important case of having computational constraints. The need for scalable computation is what made us ignore the off-diagonal elements. However, the need for better approximation made us use the exact diagonals of Hessian at the last layer since it can be computed linearly too. There are other works (e.g., AdaHessian) that have the same motivation for scalable second-order methods. We believe our work is a step in that direction.

---

> > > > > > ### Author Response · Authors · 2022-11-19
> > > > > > **Response to reviewer hQqe (2/3)**
> > > > > >
> > > > > > > (ours) We are squaring and then taking the square root, hence keeping the units correct. This typo will be fixed in our rebuttal revision paper.
> > > > > >
> > > > > > > (reviewer) Thank you for addressing this. Yet, the motivation behind “squaring and then taking the square root” is unclear. Why don’t you simply take the moving average of the diagonal Hessians?
> > > > > >
> > > > > > We can indeed take the moving average of the diagonal Hessian, and we encourage the interested reader to try this case. However, we would have an unfair comparison since other methods we compared against (e.g., Adam and AdaHessian) use the “squaring and then taking the square root” technique. This would create two differences between our optimizer and other optimizers. By making our optimizer aligned with the same technique used, we can reason about the difference in the approximation to the Hessian diagonals. If the reviewer thinks that adding more experiments with a moving average of the diagonal of Hessian would help the paper get accepted and increase its score, we are ready to run these experiments too.
> > > > > >
> > > > > > &nbsp;
> > > > > >
> > > > > > > (ours) with smaller values that are widely used (e.g., 1e-8), the importance of accuracy is large
> > > > > >
> > > > > > > (reviewer) I don’t think such a small damping value is widely used for optimization with FIM or GGN (e.g., https://arxiv.org/pdf/1806.03884.pdf, https://arxiv.org/pdf/2107.01739.pdf, https://ieeexplore.ieee.org/document/9123671).
> > > > > >
> > > > > > Other second-order methods (e.g., AdaHessian) use very small damping. We agree with the reviewer that not all methods use small damping values, and some methods use larger values. However, we used our method with a small damping value and got relatively good performance, suggesting that our Hessian diagonals approximation is useful for optimization. We agree with your original argument that with large values of damping, it’s unclear how important the approximation. This is not the case for our results. We leave the analysis for damping to future works, and we also encourage interested readers in the community to use our method in their different settings to see if there is an additional gain compared to first-order methods.
> > > > > >
> > > > > > &nbsp;
> > > > > >
> > > > > > > (ours) Our experiments show that our optimizers are outperforming Adam with a damping of 1e-8 due to the better approximation.
> > > > > >
> > > > > > > (reviewer) As I pointed out, I do not think the training results are reasonable for the optimizer comparison. As we discussed, the diagonal
> > > > > > Hessian is not necessarily a good indicator of the quality of the preconditioner. Adam’s goal is to minimize the regret in online learning, while HesScale’s goal is to estimate the diagonal elements of Hessian. Hence, it is hard to say that HesScale is “the better approximation” for training neural networks.
> > > > > >
> > > > > > We would like to differentiate between the approximation method, which we call HesScale, and its derivative optimizer, which we call AdaHesScale. The goal of Adam and AdaHesScale is the same, which is to minimize the loss. Since we use the Adam style of “squaring and then taking the square root”, we think that the only difference between our method and Adam is the use of the diagonals of the Hessian.
> > > > > > We don’t make any strong claims about the preconditioner's quality when we use the Hessians diagonals but only mention in the paper that “AdaHesScaleGN outperformed other optimizers likely due to their accurate approximation”. We think this statement means that the results only suggest this, and further experiments and work can be done to discover if better approximation can help optimization or not.
> > > > > > We would like to ask the reviewer how the statements can be more accurate and clear to increase the chances of accepting the paper.

---

> > > > > > > ### Author Response · Authors · 2022-11-19
> > > > > > > **Response to reviewer hQqe (3/3)**
> > > > > > >
> > > > > > > > As I summarized in the weaknesses, (i) the validity of HesScale’s approximation (ignoring the off-diagonal elements of Hessian w.r.t. activations) is not well-supported, (ii) the statement that “HesScale is the better approximation” is not well-supported because the comparison is not fair or the criterion is not appropriate, and (iii) the superiority of AdaHesScale over the other optimizers is not well-supported because the training results are with 12% lower accuracy than the existing baseline results.
> > > > > > >
> > > > > > > Here, we summarize our response.
> > > > > > > - For i, we argued that ignoring off-diagonal terms could indeed hurt. That’s why the Becker and LeCun 1989 paper was not widely used, although it was developed with scalability in mind. Ignoring the off-diagonal terms hurt, and this is precisely why we don’t ignore the off-diagonal element in the last layer, which increases the approximation for all earlier layers without sacrificing scalability.
> > > > > > >
> > > > > > > - For ii, we only make claims about the approximation method, HeScale, which is supported by results (Fig. 1). We don’t claim that the optimization results come from the fact that our better approximation helps the quality of the preconditioner, but they only suggest that and more work is needed to discover what improves the quality. Our method, HesScale, provides a better approximation for Hessian diagonals in neural networks. We emphasize that the applications that can potentially benefit from better approximation go beyond optimization, as we discussed in the broader impact section.
> > > > > > > Moreover, we agreed that the K-FAC comparisons should be removed from the paper, and we agree with the reviewer that only methods computing the diagonal elements should be considered (e.g., GGNMC). By removing K-FAC, we believe that our comparison would be more fair.
> > > > > > >
> > > > > > > - For iii, we clarified that the number of 12% is calculated by comparing the performance of the baseline on the train set against the performance of the results in the paper on the test set. We added a table for easier and correct comparison.
> > > > > > >
> > > > > > > We would also like to ask the reviewer what other steps would address their concerns and make the paper acceptable in their eyes. We thank the reviewer again for their helpful comments and suggestions.

---

### Official Review · Reviewer_i83K · 2022-10-26

**Confidence:** 3
**Correctness:** 4
**Technical Novelty And Significance:** 3
**Empirical Novelty And Significance:** 3
**Recommendation:** 8

**Clarity, Quality, Novelty And Reproducibility:**

The written presentation is quite clear.  There is not a lot of novelty here, but the improvement is clear; also (though the authors don't say much about it), the 1989 Becker-Lecun paper does not integrate with the Adam ideas, so there is a new combination of the Hessian approximation and modern SGD-style algorithms.

As a minor but key complaint: between the small fonts and the use of many colored lines, I found the plots very difficult to read.  This is surely made more difficult by my poor color vision, and I understand the constraints of a page limit -- but I would have been a much happier reader with something a little larger!

**Strength And Weaknesses:**

To the extent that I can read the figures, the approximation scheme does indeed seem to lead to better convergence results than competitors on the test problems.  The modification compared to B&L 89 is not large, but the authors point out that it makes a big difference in the convergence of the final method.

There is no new theory behind this, or at least none given in the paper.  The verification of the quality is purely empirical.  It would be interesting if there was a theoretical argument for why the modification makes as much difference as it does.

**Summary Of The Paper:**

The authors present a modification of a method of Becker and Lecun (1989) using a diagonal Hessian approximation to improve the convergence of stochastic gradient descent schemes.  The scheme takes about the same amount of time as standard back-propagation, unlike some other approximation schemes.  Based on this modified Hessian approximation, they introduce the AdaHessian method, which compares favorably to other standard optimizers on a set of test problems.

**Summary Of The Review:**

With a small-but-critical change to a 1989 scaling technique, together with adopting modern SGD framework, the authors introduce a new SGD-style optimizer for NN training that includes second-order information "for cheap" and seems to lead to better training performance (in terms of test accuracy vs time) than natural competitors.  I am a fan of "not a big change, but the right change" work, and would like to see this published.

---

> ### Author Response · Authors · 2022-11-12
> **Thank you for your review**
>
> Thank you for your valuable feedback. Please let us know if you have any more questions.
>
> > It would be interesting if there was a theoretical argument for why the modification makes as much difference as it does.
>
> We can provide an intuition for why our modification makes much difference. Using the Hessian diagonals propagation equation (Eq. 4), we see that the Hessian diagonals at the $l$-th layer are a function of the Hessian diagonals at the $l+1$ layer and the Hessian diagonals at the $l+1$ depend on the elements from the next layer. This means any better approximation at the last layer $L$ will improve the approximation at all layers.
>
> > between the small fonts and the use of many colored lines, I found the plots very difficult to read. This is surely made more difficult by my poor color vision, and I understand the constraints of a page limit -- but I would have been a much happier reader with something a little larger!
>
> Thank you for letting us know. We will try our best to use larger fonts in all figures without exceeding the page limit.
>
>
> We would like to thank you again for reviewing our paper!

---

### Author Response · Authors · 2022-11-15
**Rebuttal Revision**

We would like to let all reviewers know that we uploaded a rebuttal revision, incorporating the valuable feedback we received. Thank you all for your reviews!

---

### Decision · Program_Chairs · 2023-01-20

**Decision:**

Reject

**Justification For Why Not Higher Score:**

Limited evaluation, does not perform better than ADAM, contribution is oversold

**Justification For Why Not Lower Score:**

N/A

**Metareview: Summary, Strengths And Weaknesses:**

This paper proposes a method to approximate the diagonal of Hessians, based on a simple modification of the method of Becker and Lecun (1989).

Strengths:
- The method is simple
- It seems to work well on MNIST and CIFAR

Weaknesses:
- The paper is overselling its contribution: It is not mentioned until Section 3 that the proposed method is a simple modification of Becker and Lecun (1989)
- No theoretical guarantees
- Experiments only on MNIST and CIFAR, on feedforward and CNN networks: for a strictly empirical paper, this is too limited
- Does not perform better than ADAM
- Figure 1 is misleading, as there is no evidence that approximating the Hessian diagonal well is useful for optimization

Overall, we agree with the 2 reviewers who recommended rejection. The paper does not meet the bar for publication at ICLR, in its current state.